# Silicon Isotope Geochemistry: Fractionation Linked to Silicon Complexations and Its Geological Applications

**DOI:** 10.3390/molecules24071415

**Published:** 2019-04-10

**Authors:** Wei Wang, Hai-Zhen Wei, Shao-Yong Jiang, Xi Liu, Fang Lei, Yi-Bo Lin, Yao Zhao

**Affiliations:** 1State Key Laboratory of Geological Processes and Mineral Resources, School of Earth Resources, China University of Geosciences, Wuhan 430074, China; cugweiwang@163.com (W.W.); eeyz@leeds.ac.uk (Y.Z.); 2State Key Laboratory for Mineral Deposits Research, School of Earth Sciences and Engineering, Nanjing University, Nanjing 210023, China; liuxi069@163.com (X.L.); yibolin@smail.nju.edu.cn (Y.-B.L.); 3CAS Center for Excellence in Comparative Planetology, China, Anhui 230026, China; 4School of Geographic and Oceanographic Sciences, Nanjing University, Nanjing 210023, China; leifang715@163.com

**Keywords:** silicon isotopes, equilibrium isotope fractionation, dynamic isotope fractionation, the Rayleigh fractionation model, speciation/coordination of silicon

## Abstract

The fundamental advances in silicon isotope geochemistry have been systematically demonstrated in this work. Firstly, the continuous modifications in analytical approaches and the silicon isotope variations in major reservoirs and geological processes have been briefly introduced. Secondly, the silicon isotope fractionation linked to silicon complexation/coordination and thermodynamic conditions have been extensively stressed, including silicate minerals with variable structures and chemical compositions, silica precipitation and diagenesis, chemical weathering of crustal surface silicate rocks, biological uptake, global oceanic Si cycle, etc. Finally, the relevant geological implications for meteorites and planetary core formation, ore deposits formation, hydrothermal fluids activities, and silicon cycling in hydrosphere have been summarized. Compared to the thermodynamic isotope fractionation of silicon associated with high-temperature processes, that in low-temperature geological processes is much more significant (e.g., chemical weathering, biogenic/non-biogenic precipitation, biological uptake, adsorption, etc.). The equilibrium silicon isotope fractionation during the mantle-core differentiation resulted in the observed heavy isotope composition of the bulk silicate Earth (BSE). The equilibrium fractionation of silicon isotopes among silicate minerals are sensitive to the Si–O bond length, Si coordination numbers (CN), the polymerization degrees of silicate unites, and the electronegativity of cations in minerals. The preferential enrichment of different speciation of dissoluble Si (DSi) (e.g., silicic acid H_4_SiO_4_^0^ (H4) and H_3_SiO_4_^−^ (H3)) in silica precipitation and diagenesis, and chemical weathering, lead to predominately positive Si isotope signatures in continental surface waters, in which the dynamic fractionation of silicon isotope could be well described by the Rayleigh fractionation model. The role of complexation in biological fractionations of silicon isotopes is more complicated, likely involving several enzymatic processes and active transport proteins. The integrated understanding greatly strengthens the potential of δ^30^Si proxy for reconstructing the paleo terrestrial and oceanic environments, and exploring the meteorites and planetary core formation, as well as constraining ore deposits and hydrothermal fluid activity.

## 1. Introduction

As silicon is the third most abundant element on Earth (~16.1 wt.% Si) [1], the most abundant non-volatile element in the solar system after oxygen, and the second most abundant element in the upper crust (28.8 wt.%) [2], it is expected to be one of most suitable elements for understanding the fundamentals of various processes in Earth [3]. Silicon has three stable isotopes with the following abundances: ^28^Si: 92.27%, ^29^Si: 4.68%, ^30^Si: 3.06% [4], and silicon isotope compositions are reported as δ^30^Si or δ^29^Si in per mil (‰), defined as the deviation from the silicon isotope standard reference material (NBS-28) as follows (Equations (1) and (2)):δ^30^Si = 1000{[(^30^Si/^28^Si) _sample_/(^30^Si/^28^Si) _standard_] − 1}(1)
δ^29^Si = 1000{[(^29^Si/^28^Si) _sample_/(^29^Si/^28^Si) _standard_] − 1}(2)

The silicon isotope fractionation (α_A−B_) between the two substances A and B is defined as:(3)αA−B=(Si30/Si28)A(Si30/Si28)B.

According to the isotopic differences between substances A and B (Δ^30^Si_A−B_), the approximation of Δ^30^Si in term of silicon isotope fractionation factor α_A−B_ can be made below:Δ^30^Si_A−B_ = δ^30^Si_A_ − δ^30^Si_B_ ≅ 1000(α_A−B_ − 1) ≅ 1000lnα_A−B._(4)

Because of the large budget of the mantle Si reservoir, and the long history of convective mixing [3], it has been widely recognized that the thermodynamic isotope fractionation of silicon associated with high-temperature processes is smaller than low-temperature processes, e.g., planetary formation or solar nebular processes [5,6,7,8] and magmatic–metamorphic rock formation [9,10]. By contrast, significant isotope fractionation of silicon, with δ^30^Si ranging from −5.7‰ to +6.1‰, occurs during low-temperature geochemical cycling, including the soil processes (i.e., weathering of rocks) [11,12,13], and biogenic silica formation by plants (both opal from diatoms and phytoliths from terrestrial plants) [9,14,15,16,17,18,19,20,21,22,23,24,25]. Such effects provide powerful geochemical constraints for tracing bio-physico-chemical processes in terrestrial environments [10,11,12,13,26,27,28,29,30] and mineral deposits formation [31,32,33,34,35,36,37].

As early as 1924, the initial atomic weight of silicon was given by Jaeger and Dijkstra according to the silicon isotopic compositions of Earth’s rock samples (e.g., granite, volcanic rock, etc.) and stone meteorites [38]. After that, there were progressive advances in the silicon isotope geochemistry since the 1950s. For instance, the silicon isotope compositions of minerals and rocks were firstly reported by Reynolds and Verhoogen [39], and the fractionation mechanism of silicon isotope in the nature was provided by Allenby and Grant [40,41]. The earliest report on the variations of ^30^Si/^28^Si ratios in lunar rocks was provided by Epstein and Taylor [42], and those in both the moon samples and the meteorites were investigated by Clayton [43]. In the early 1990s, a higher analytical precision of <±0.1‰ for the silicon isotope analysis was obtained using gas source mass spectrometry (GS-MS) [10,15,16,44,45,46,47]. Most recently, a robust, fast, and reliable protocol to determine silicon isotope ratios using multicollector inductively plasma mass spectrometry (MC-ICP-MS), was established by Georg et al. [48], archiving the highest precision in various geological materials (±0.07‰, 2σ). On the basis of that, it has become possible to better explore minor isotopic fractionations in nature, such as the silicon isotope geochemistry in biosphere (e.g., References [13,17,18]), hydrosphere (e.g., References [18,49,50,51,52]), meteorites and planetary core formation [7,50,53,54,55,56,57], etc.

During the initial study period from the 1950s to 1990s, the progresses of silicon isotope geochemical studies have been well demonstrated by Ding et al. [10]. The recent understandings of silicon isotope geochemistry in high-temperature processes have been systematically summarized by Savage et al. [58], including the silicon isotope compositions in Earth’s accretion and core formation and continental crust, silicon isotope fractionation behavior during magmatic differentiation, etc. Additionally, a comprehensive review of silicon isotope geochemistry was given by Poitrasson et al. [59], covering elemental and isotopic abundances in extraterrestrial and terrestrial reservoirs, as well as the elemental and isotopic behaviors during major geological processes and relevant implications. The continental Si cycle and its impact on the ocean Si isotope budget have been well stressed by Frings et al. [60]. The influence of the evolution of biosilicifying organisms on oceanic dissolved Si (i.e., DSi) inventory since the beginning of oxygenic photosynthesis and its implication for the cycling of carbon and other key nutrients in the ocean were demonstrated by Conley et al. [61]. The recent contribution by Sutton et al. highlighted the latest understanding of Si cycle in marine, atmospheric, freshwater, and terrestrial systems and emphasized on existing problems (e.g., high-quality δ^30^Si data, multiple bio-geochemical proxies, and parameters) and potential challenges (e.g., controls on silicon isotope fractionations) for silicon isotope geochemistry studies [62]. In this work, the fundamental advances in analytical approaches, the silicon isotope fractionation linked to silicon complexation/coordination and thermodynamic conditions in various geological processes (e.g., silicate minerals with variable structures and chemical compositions, silica precipitation and diagenesis, chemical weathering of crustal surface silicate rocks, biological uptake, and the global oceanic Si cycle), and the geological implications have been summarized in detail, in an attempt to enhance fundamental understanding of the silicon isotope geochemical constraints for tracing bio-physico-chemical processes in terrestrial, oceanic environments, and the Earth’s interior processes.

## 2. Analytical Techniques

In early studies, the silicon isotope compositions were measured by using the gas-source mass spectrometry (GS-MS), whereby the silicon was analyzed in the form of the volatile SiF_4_^+^ [10]. Then, the in-situ analysis of silicon isotopes has been achievable with the secondary ion mass spectroscopy (SIMS) since 1980s [63,64,65]. More recent development of multicollector inductively plasma mass spectrometry (MC-ICP-MS) opens the new era of silicon isotope geochemistry in geological, environmental, and material science in recent decades, attributing to low sample-size requirements, high sample throughput, and improved precision and sensitivity [66].

### 2.1. Gas-Source Mass Spectrometry (GS-MS)

For the analysis of silicon isotope compositions using the gas-source mass spectrometry (GS-MS), there are four methods for transforming solid silicates into gaseous SiF_4_. The first one is to digest the solid rocks/minerals in hydrofluoric acid (HF), followed by precipitation with BaCl_2_ as BaSiF_6_. The precipitate is thermally decomposed into gaseous SiF_4_ to transfer to the GS-MS [39]. With the second approach, the solid material is directly transformed into SiF_4_ by adding the mixture gas of F_2_ and HF as the fluorinating agent [42]. The third one is to transform the silicate materials into gaseous SiF_4_ by using the hazardous chemicals of BrF_5_ or F_2_ [67], which has been extensively employed in relevant laboratories. After coupling pretreatment procedures to remove any impurities from siliceous rocks, the external reproducibility of the SiF_4_-GS-MS approach is as high as ±0.1‰ (2σ) [10]. Fourthly, the most recent approach is to transform solid samples into gaseous SiF_4_ by using the acid decomposition of Cs_2_SiF_6_, providing an analytical precision generally better than ±0.15‰ (2σ) [68]. However, the further application of SiF_4_-GS-MS approach is greatly limited because of obvious disadvantages, such as complicated sample preparation procedure, time-consuming nature, and use of extremely hazardous and fluorine-based chemicals [16,44].

### 2.2. Second Ion Mass Spectrometry (SIMS)

The in situ analysis of silicon isotopes has been achievable with the secondary ion mass spectroscopy (SIMS) since the 1980s [63,64,65,69]. The obvious advantage of high spatial resolution enables it to investigate silicon isotope compositions in chert [14,70] and in banded iron formations (BIFs, [71]). The method is relatively limited for wider application because of the limited external reproducibility from ±0.2‰ to ±0.3‰ (2σ) compared to the solution MC-ICP-MS approach.

### 2.3. Multicollector Inductively Plasma Mass Spectrometry (MC-ICP-MS)

More recent development of multicollector inductively plasma mass spectrometry (MC-ICP-MS) opens a variety of applications in geological, environmental, and material science where both high spatial resolution and accuracy are required [66]. The digestion of silicate materials using hydrofluoric acid (HF) would induce a series of analytical problems, such as special equipment for HF-resistant systems and artificial isotope fractionation caused from gaseous silicon loss. A new technique to separate and purify Si based on alkaline fusion followed by ion-exchange chromatography has been provided by Georg et al. [48], making the measurement of silicon isotope more convenient, faster, and reliable. With this modification, the high-resolution mass-spectrometry can avoid the polyatomic interferences (e.g., ^14^N^16^O^+^), and the recoveries >98% indicate that silicon losses through polymerization are insignificant. This technique is able to determine relative Si isotope variations of samples with a long-term external reproducibility of ±0.14‰ (2σ) [48,72]. Meanwhile, the problems of being sensitive to matrix effects, mass discrimination, and temporal drift in isotopic ratios still remain great challenges for high-precision Si isotope analysis. One of the disadvantages of this method is the often non-quantitative removal of the sample matrix such as anions (e.g., SO_4_^2−^) and dissolved organic carbon (DOC), which induce shifts in the silicon isotope composition. An improved procedure is to remove SO_4_^2−^ by forming barite precipitation and to remove DOC by the combined action of UV-C and ozone [73,74]. The effects of other major anionic species (e.g., PO_4_^3−^, B(OH)_4_^−^, and Cl^−^) on silicon isotope values still need to be investigated.

### 2.4. Laser Ablation Multicollector Inductively Coupled Plasma Mass Spectrometry (fsLA-MC-ICP-MS)

An analytical protocol for accurate in situ Si isotope analysis has been established on a new second-generation custom-built femtosecond laser ablation system. The laser is coupled to a multicollector inductively coupled plasma mass spectrometer (fsLA-MC-ICP-MS) [75,76]. To resolve Si isotope signals from isobaric interferences (mainly ^14^N^16^O polyatomic ions), the ion optics are operated in medium mass resolution mode, with which an external reproducibility better than ±0.23‰ (2σ) can be obtained. It is suitable to investigate the Si isotope signature of rock weathering at the micro-scale.

## 3. Silicon Isotope Variations in Major Reservoirs and Geological Processes

The silicon isotope compositions (δ^30^Si) relative to the NBS-28 reference standard in the bulk silicate Earth, bulk Moon, and bulk silicate Mars are −0.29 ± 0.07‰, −0.27 ± 0.04‰, and −0.49 ± 0.03‰. [59]. The δ^30^Si in continental crust shows a narrow range from −0.43‰ to −0.15‰ with the relationship of δ^30^Si = 0.0056 × SiO_2_ (wt%) −0.567 [57], and that in the oceanic crust, upper mantle, and ocean are estimated to be −0.36 to −0.22‰, −0.39 to −0.23‰, and +0.5 to +4.4‰ [59]. The δ^30^Si of −0.38‰ to −0.27‰ with an average value of −0.32 ± 0.06‰ (2SD) for the altered oceanic crust is given by Yu et al. [77], in good agreement with the previous estimation [58]. The general trend of the δ^30^Si variations in various geological processes in terrestrial reservoirs is given in Figure 1. It is clear that the thermodynamic isotope fractionation of silicon associated with high-temperature processes is smaller than that in the low-temperature geological processes, such as chemical weathering, biogenic/non-biogenic precipitation, adsorption, and biological uptake. With further understanding on the driving mechanisms for silicon isotope fractionations behind typical geological processes, the silicon isotope geochemistry provides powerful geochemical constraints on tracing bio-physico-chemical processes in terrestrial environments, formation of mineral deposits, hydrothermal fluids activities, and meteorite and planetary evolutions, etc., which will be discussed in detail below.

## 4. Silicon Isotope Fractionations Linked to Silicon Coordination/Complexation

### 4.1. Silicate Minerals with Variable Structures and Chemical Compositions

Despite silicon being the defining element of silicate reservoirs on Earth, there is still no clear understanding of how much is hosted in Earth’s core or how the silicon-enriched continental crust forms from a long history of weathering, erosion, and subduction [58]. Like that of any other elements, silicon isotope fractionation is relatively limited at high-temperature processes [79]. The potential for isotopic fractionation is further limited by the low volatility, single valence state, and invariant bonding environment. The silicon isotope fractionation behavior during magmatic differentiation has been systematically summarized by Savage et al. [58]. In addition to the experimental approach, the quantum mechanical calculation (e.g., the first-principles calculation based on density functional theory) shows a powerful capacity for calculating equilibrium fractionation factors of silicon isotopes in high temperature processes in Earth’s deep interior. The comprehensive studies indicated that the equilibrium fractionation of silicon isotopes is sensitive to the bond strength (Si–O) with the heavier isotopes preferring the stronger bond (e.g., phase transformations among various minerals) and variation in Si coordination numbers (CN), and Si isotopes can be significantly fractionated among minerals with different Si CNs, e.g., between Mg-perovskite (CN=6) and olivine polymorphs (CN = 4) and between ^VI^Si and ^IV^Si in majorite. In olivine polymorphs, olivine is slightly enriched in the heavy Si isotope compared to wadsleyite and ringwoodite due to their different crystal structures [80,81] (Figure 2, the initial crystal structures were obtained from American Mineralogist Crystal Structure Database, http:// rruff.geo.arizona.edu/AMS/amcsd.php).

The initial theoretical understanding of equilibrium silicon fractionation between quartz and kaolinite was given by Méheut et al. (2007) [82]. With various polymerization degrees of silicate units *Q^n^* (where n denotes the number of bridging oxygens for one SiO_4_ unit), the equilibrium silicon isotope fractionation properties of quartz (*Q*^4^), lizardite (*Q*^3^), kaolinite (*Q*^3^), and enstatite (*Q*^2^), and forsterite (*Q*^0^) were investigated by Méheut et al. (2009) [83], which revealed out that the equilibrium silicon isotope fractionation trend is not directly connected to the polymerization degree. Kaolinite and lizardite, with the same polymerization degree, have very different fractionation properties, suggesting that other cations, in particular Al, can play a significant role in determining the isotopic fractionation properties of silicon. In addition, silicate minerals with similar chemical compositions (i.e., muscovite, KAl_2_Si_3_AlO_10_(OH)_2_) constituted of TOT layers (T being a tetrahedral Si_3_Al layer, O being an octahedral aluminous layer), kaolinite (Al_2_Si_2_O_5_(OH)_4_) constituted of TO layers) have shown the similar silicon isotopic fractionation properties. The most recent understanding proposed that the silicon isotope fractionation properties among phyllosilicates (kaolinite, lizardite, pyrophyllite, talc) appear to be correlated with stoichiometry (Equation (5)) [84]. It indicates that the effect of cation X on silicon isotope fractionation increases with decreasing electronegativity of X, and explains the enrichment in heavy silicon isotopes accompanying magmatic differentiation [84].
(5)$$1000ln\alpha^{30}Si_{phyllosilicate-quartz}\;=\;\frac{\alpha_{Mg}\left(T\right)\cdot Mg^{eq.}+\alpha_{Al}\left(T\right)\cdot Al^{eq.}}{Si^{eq.}}$$
where Si^eq^ = #Si, Al^eq^ = 3/4#Al, and Mg^eq^ = 1/2#Mg (cation equivalents) are the charge-weighted stoichiometric coefficients of each cation, normalized to the charge of the silicon atom, and α_X_(T) are proportionality coefficients depending on temperature.

### 4.2. Silica Precipitation and Diagenesis

The most common species of Si in solution at ambient conditions are silicic acid H_4_SiO_4_^0^ (H4), and its associated base H_3_SiO_4_^−^ (H3) and H_2_SiO_4_^2−^ (H2), when the Si concentration is below the solubility limit of amorphous silica (i.e., 1.93 mmol/L Si at 25 °C and pH ~8) [85]. The fraction of each species H4, H3, and H2 with solution pH from 7 to 12 is given in Figure 3b according to the dissociation constants of pK_a1_ = 9.84 and pK_a2_ = 13.2 at 298 K and 0 M ionic strength condition [86]. The equilibrium silicon isotope fractionation between H_4_SiO_4_^0^ and H_3_SiO_4_^−^ species has been calculated using the first-principles methods based on ab initio molecular dynamics simulation, with which the fractionation factor (α_H3-H4_) of 0.9984 ± 0.0003 (i.e., Δ_H3-H4_ = −1.6 ± 0.3‰) was obtained at 300K [87], as shown in Figure 3b. It implies that H3 enriches the light isotope while H4 enriches the heavy one (Figure 3a), and also suggests the important impact of speciation on silicon isotope fractionation in particular pH [87]. The silicon isotope fractionation in protonation of H4 was further investigated both theoretically and experimentally by Fujii et al. [86]. With the dataset (Table 2 within Fujii et al. [86], the α_H3-H4_ values of 0.9963 ± 0.0009 and 0.9981 ± 0.0005 are derived from the isolated H_4_SiO_4_ and H_3_SiO_4_^−^ species and hydrated species H_4_SiO_4_⋅(H_2_O)_m_ and H_3_SiO_4_^−^⋅(H_2_O)_n_ (m = 8; n = 1, 7, 9). Clearly, the consistence in α_H3-H4_ value of 0.9981–0.9984 from hydrated species from individual studies [86,87] verifies the influence of the solvation layer on fractionation and it might represent the real silicon isotope fractionation property in solution. Considering the molar fraction of H_2_SiO_4_^2−^ (H2) is less than 0.59% when solution pH ≤ 11 (Figure 3a), the δ^30^Si of H4 and H3 species relative to the bulk solution (δ^30^Si = 0) with α_H3-H4_ values of 0.9963, 0.9972, 0.9981, and 0.9984 as a function of pH are estimated in Figure 3b, with reference to the previous study by Fujii et al. [86]. As a result, a new paleo-pH proxy using silicon isotope compositions in seawater or seawater-derived fluids and precipitated silica was proposed by Fujii et al. [86], which extend the applicable pH range to 9–12 compared to the boron isotope paleo-pH proxy (pH 7.8–10) [88].

In surficial geological processes, the occurrence of polymerized silicic acid (PS) might be caused by the weathering of silicate liberating from Al–Si-containing solids and polymerization of silicic acids in soil-liberating H_2_O molecules during cyclic freezing or evaporation [89], which has an impact on the understanding of the dissolution mechanism of aluminosilicates in soils, sediments, and rocks. The activity of Al^3+^ and H_4_SiO_4_^0^ on a stability diagram was provided using speciation calculation, in which a positive relationship between the occurrence of PS and the saturation degree was identified, suggesting polymerization would be precursors for the formation of secondary Al-silicates [89]. In marine environments, biogenic silica is primarily produced by planktonic organisms living in the surface ocean and a large fraction of biogenic silica is recycled via dissolution within the upper 100 m of the water column [90]. Generally, the equilibrium solubility of Si increases with increases in environmental temperatures, and is highly variable with variations in temperature, pressure, and differences in specific surface area and Al content of bio-siliceous fragments [91]. In fluid phase, precipitation of silica in excess of the equilibrium solubility is initial through nucleation of colloidal amorphous silica particles (via combination of silicic acid molecules, H_4_SiO_4_ + H_4_SiO_4_ = H_6_Si_2_O_7_ + H_2_O, etc.) and subsequently through growth of the nucleation of colloidal amorphous silica particles via continuing addition of silicic acid to the existing surfaces (i.e., H_4_SiO_4_ + HO-(SiO_2_)_n_ = (SiO_2_)_n_-O-Si(OH)_3_ + H_2_O) [92]. In addition to dissolved Si concentration, other factors, such as temperature, pH, OH^-^, Na^+^, and Al^3+^, also have an impact on silica precipitation and phase transformation [93,94,95,96,97,98,99]. The most important parameter may be the incorporation of Al in the precipitating phase, resulting in surface defect/surface nucleation-controlled precipitation [99]. Aluminum decreases silica solubility, and its incorporation in the precipitating phase can cause surface defects and silicon isotope fractionation [29,99,100]. Considering the silicic acid does not dissociate until exposed to distinctly alkaline solution and that hydroxyl ion is known to catalyze the polymerization of silicic acid [101], it was proposed to represent the dissociation of aqueous silica as a neutralization rather than ionization reaction (Equation (6) [87]). Using the neutralization reaction and neutral-pH solubility, the solubility of silica polymorphs as a function of temperature and pH in circumstances was expressed as below (Equation (7) [87]).
(6)$$H_{4}SiO_{4}+OH^{-}=H_{3}SiO_{4}^{-}+H_{2}O$$
(7)$$S=K_{c}^{\prime }\left[1+\frac{10^{pH}K_{B\;}K_{W}}{\gamma_{H_{3}SiO_{4}^{-}}}\right]$$
(8)$$logK_{c}^{\prime }=0.338+\frac{840.1}{T}-7.889\times 10^{-4}T$$
where K_B_ and K_W_ are the thermal equilibrium constant for the dissociation of aqueous silica (Equation (6)) and the dissociation of H_2_O; *K_c_*′ is the silica solubility in water (neutral pH, M); γ is the activity coefficient of silicate species H_3_SiO_4_^−^; S is the total solubility of silica under different pH and T conditions.

As shown in Figure 4a, silica solubility increases exponentially with pH (especially above pH 8) and temperature, and acidification might control silica deposition and polymerization in geothermal fluids. Precipitation of silica from solution is governed by silicic acid equilibrium relative to amorphous silica [92]. The equilibrium silicon isotope fractionations of quartz-H_4_SiO_4_ and kaolinite–H_4_SiO_4_ were calculated to be α_quartz-H4_ of 1.0021 (i.e., Δ_quartz-H4_ = +2.1 ± 0.2‰) and α_kaolinite-H4_ of 1.0004 (i.e., Δ _kaolinite__-H4_ = +0.4 ± 0.2‰) [87] (Figure 4b). The results are irreconcilable with natural observations of fractionations during silica or clay precipitation, suggesting that these fractionation processes obey kinetic laws instead. The dynamic fractionation of silicon isotope during precipitation of silica gel from solution was studied experimentally [99]. As shown in Figure 4c, this precipitation process of silica was well described by the Rayleigh fractionation model, and the calculated kinetic fractionation factors of α_precipitated-dissolved_ range from 0.9990 to 0.9996 at room temperature ranges, which confirms the preferential deposition of ^28^Si during abiotic silica precipitation [102,103]. Similar preferential enrichments of lighter isotopes in the solid phase also have been observed, such as for Fe-isotopes during rapid precipitation of hematite [104], Ca-isotopes during calcite precipitation [105], and Mg-isotopes in magnesite deposition [105].

Precipitation of silica is a polymerization process of monosilicate (i.e., H_4_SiO_4_) in solution, and the bond of ^28^Si–O is weaker than the bond of ^30^Si–O. The light silicon isotope is always preferentially taken up in polymerization and precipitation, which leads to dissolved silicon including less ^28^Si than precipitated silicon [108]. During the non-biological black chert formation, the kinetic fractionation factors ranging from 0.9965 at 10 °C and 0.9993 at 35 °C under pH 8.5 are larger than those of equilibrium isotope effects from 0.9995 at 10 °C to 1.0005 at 35 °C. It indicates that silicon isotope fractionation is strongly dependent on the degree of (metastable) equilibrium in the silica–water system. Large isotope fractionation occurred in the silica–water system of natural systems where dissolved and precipitated silica is not equilibrated [37]. In accordance with this mechanism, two distinctive silicon isotope fractionation behaviors were well interpreted: A smaller isotope fractionation during the silicification of modern sandstone deposits or Archean silicified volcanic-sedimentary rocks (S-chert) implied that the slow percolation of siliceous fluids through pore spaces reached equilibrium between dissolved and precipitated silica. For instance, the δ^30^Si value of +1.1‰ in S-chert is almost identical to that of ~ +1‰ in Archean seawater [53,71,109,110]; a larger isotope fractionation observed in modern geothermal sinter deposits (down to −4.0‰) and Archean orthochemical cherts (down to −4.3‰) can be explained by rapid precipitation of silica from oversaturated solutions dominated by kinetic isotope fractionation [37]. Besides, the results are consistent with the observation that silicon isotope fractionation during silica precipitation is temperature dependent and more significant at low temperatures [37,102,103]. The effect of Al on the silicon isotope fractionation in experiments with continuous precipitation and dissolution of silica indicates that the enrichment of light Si isotope found in natural environments is caused exclusively by a unidirectional kinetic isotope effect during fast precipitation of solids, aided by co-precipitation with Al phases or other carrier phases (e.g., Fe(III)) because of adsorption or binding of silicon onto Al-hydroxide [111] and Fe-oxide [112]. By contrast, during slow precipitation, or in the absence of a carrier phase like Al, no Si isotope fractionation is observed, representing the equilibrium isotope fractionation of silica precipitation [30,111].

### 4.3. Chemical Weathering of Crustal Surface Silicate Rocks

During chemical weathering processes (Reaction I), silicon is either being released into continental surface and ground waters, or transported into soils by uptake of plants, and formation of secondary precipitates (e.g., metastable silica-containing solids), as well as adsorption onto secondary oxides. The associated dynamic fractionation occurs in precipitation during the weathering of silicate minerals, where the lighter silicon isotopes combines with Al and Fe-oxides to form the clay precipitation and the heavy silicon isotopes enrich in fluid phases or in secondary precipitation [46,71]. The mechanism of silicon adsorption by Al and Fe-oxides was explained as the interaction of H_4_SiO_4_^0^ with surface -OH groups to form silicate bi- den-date innersphere complex [113,114], accompanying surface Si polymerization [113,114,115]. Results from the substitution of silica for double-corner FeO_6_-octahedra in iron oxy-hydroxide polymeric complexes existing at the early stages of Fe(III) hydrolysis show that silica substitutes for double-corner, likely by forming 2C-type (double corner) complexes with small Fe oxy-hydroxide polymers whose structure consists of FeO_6_-octahedra linked together by common edges [113]. The silicon isotope fractionation during the specific adsorption of monosilicic acid by iron minerals (ferrihydrite, goethite) fitted better a Rayleigh distillation path, and the fractionation factors were determined to be α_mineral-solution_ of 0.9990 (i.e., Δ^30^Si_mineral-solution_ = −1.04 ± 0.03‰) and 0.9984 (i.e., Δ^30^Si _mineral-solution_ = −1.56 ± 0.12‰) for ferrihydrite and goethite, respectively [112]. On the basis of the adsorption and polymerization behavior of H_4_SiO_4_^0^ on ferrihydrite [115], and the X-ray absorption fine structure spectroscopy study in iron (III)-silica interaction [113], we illustrate the bonding structures of polymerized silicic acid with ferrihydrite and goethite schematically as shown in Figure 5. The silicon isotope fractionation properties clearly indicate the preferential attachment of H_3_SiO_4_^−^ (H3) over H_4_SiO_4_^0^ (H4) species with reference to silicon isotope fractionation between H4 and H3 (Figure 3), which might be attributed to the variation on solution pH occurred in Fe(III) hydrolysis in the presence of silica or static attraction of negatively charegd H3 with charged iron oxy-hydroxide minerals and colloids. The study on the impact of H_4_SiO_4_^0^ adsorption on the fractionation of silicon isotope in basaltic ash soil showed that the light Si isotope preferentially adsorbed onto soil Fe-oxides, and the enrichment of heavy isotope in solution increased with increasing of the extent of soil weathering degree, iron oxide content, and proportion of short-range ordered Fe-oxide [116]. This result provided direct experimental evidence that the adsorption could contribute to the sequestration of light Si isotopes in clay-size fractions of soils [116]. As the local streams were fed by pore water from weathering profiles, the heavy silicon isotopic riverine signature characterized the hydrological output of silicon compared to that of crustal rocks [49,50,116]. The integrated effects on silicon isotope distributions were investigated along climate gradients of basalt-derived soil in Hawaii, identifying that four distinct processes could control the bulk δ^30^Si of basalt-derived soil: Primary mineral weathering, secondary mineral weathering, mineral dust accumulation, and bio-cycling [117]. The associated isotope fractionation effects would be summarized as: (i) Mass-dependent fractionation of silicon isotopes generating more negative values of δ^30^Si in secondary soil minerals and higher values of δ^30^Si in soil water compared to the primary mineral Si source; (ii) climate condition has indirect effects on soil δ^30^Si along a gradient of basalt-derived soils with the annual rainfall of 160–2500 mm, and bulk soil δ^30^Si declined in drier climate and increased in wetter climate soils with Si depletion; (iii) more negative δ^30^Si values in upper soil horizons are likely the result of additional fractionation that occurs during repeated dissolution and precipitation of secondary phases, while δ^30^Si values of soil horizons >1 m deep were better correlated with Si depletion [117]. Reaction I:2Na(K)AlSi_3_O_8_ + 2CO_2_ + 3H_2_O → Al_2_Si_2_O_5_(OH)_4_ + 2Na^+^(K^+^) + 2HCO_3_^+^ + 4SiO_2_

With the prevalence of silicon isotope as a useful weathering proxy, it raised the question of the role of mineralogy on silicon isotope fractionation by silicon incorporation into secondary phases. It is commonly accepted that the Si content in soils decreases progressively with increasing degree of weathering following preferential incorporation of light isotopes in secondary clay minerals (such as References [26,118]), while it also was suggested that limited silicon mobility preserves Si isotope compositions in soils close to that of the parent silicate material [119]. In order to understand the silicon depletion in soils and the influence of clay mineralogy on silicon isotope fractionation, the δ^30^Si values in bulk soil and clay fractions in three weathering sequences at Guadeloupe formed under contrastive climatic conditions were compared by Opfergelt et al. [12]. This study clearly indicated that the silicon isotope composition in secondary clay minerals was controlled by the degree of soil desilication, which was related to the rainfall pattern governing the formation of second minerals. Silicon isotope fractionation between the parent silicate material and the secondary clay minerals increases following the order of smectite to allophane to halloysite to kaolinite, which corresponds to a smaller isotope fractionation for Si-rich clay minerals (e.g., smectite) and a larger isotope fractionation for Si-poor clay minerals such as kaolinite [13]. It explained the variation trend of δ^30^Si in secondary clay minerals, such as slightly lighter δ^30^Si in smectite (−0.16 to −0.52‰) [51] and very much lighter in kaolinite (−2.2‰) [26,117], compared to the limited spread of δ^30^Si (i.e., −0.29 ± 0.08‰) in silicate rocks (BSE) [3,57,120].

In summary, secondary minerals formed with high Al/Si ratios are generally depleted in ^30^Si (see References [120,121]), and those in river water are enriched in ^30^Si over the host rock (e.g., References [49,116,117,118,119]). In addition, the δ^30^Si values show an upward trend from upstream to downstream [4]. The associated silicon isotope fractionation during the crustal surficial weathering is schematically shown in Figure 6. The integrated understanding greatly strengthens the potential use of δ^30^Si proxy for the paleo-reconstruction of climatic conditions in soil weathering environments.

### 4.4. Biological Uptake

Silicon is involved in biological processes via the uptake of silicic acid by plant tissues and formation of deposits of amorphous silica (e.g., phytoliths), from which it can be released into soils and fluids by physical and chemical weathering and re-deposited in soils as secondary silicate minerals such as allophanes and imogolites or be exported to aquatic ecosystems [122,123]. The contribution of plants to biogeochemical Si cycles related to weathering processes is significant because of larger biologic turnover of silicon. For instance, the contribution of plants to biogeochemical Si is as high as 58–76 kg/ha/y in the equatorial forest ecosystems [124]. Therefore, the uptake, storage, and release of silicon by the vegetation have to be taken into account when using dissolved silicon for tracing chemical weathering dynamics. Currently, there are different opinions on the issue of how roots take up and transport silicic acid, e.g., passive diffusion after active root uptake and transport for oats [125], selective absorption of silicate particle for typical plants [126], as well as silicate–organic compounds (e.g., silicate–sugar complexes in our previous contribution [78]). During these processes, the aqueous silicate solutions could react with straight-chain polyhydroxy compounds to form stable hypervalent silicon complexes [78,127,128,129,130,131]. Recent contributions clarified a series of stable organo-silicate complexes which form in aqueous alkaline silicate solutions upon addition of certain aliphatic polyols (such as mannitol, xylitol, and threitol) [131], and sugar acids (such as gluconic, saccharic, and glucoheptonic acids) [78,132]. In these complexes, silicon is tetracoordinated, hexacoordianted, or exists as a hypervalent pentaoxo or hexaoxo centre (Figure 7) [78,127,128,129,130,131,132]. It turns out that Si–O–C bonding is actually quite common in these processes. The apparent readiness to combine in aqueous solution indicates that sugar acids, or bio-molecules containing sugar acid substructures, play an important and hitherto unrecognized role in the geochemistry and biochemistry of silicon. Further, it was reported that some enzymes, silaffin, and polyamine chains were capable of polymerizing silicic acid at neutral to acidic pH conditions [131,132,133], but it is questionable whether these compounds exist in plants [15].

The role of complexation in biological fractionations of silicon isotopes is very poorly understood, largely because the complicated biochemical pathways, likely involving several enzymatic processes and active transport proteins, are not well constrained in any siliceous organism. Further, it was found that silicon isotope fractionations between plants and growth solutions were basically similar in both direction and extent for rice, bamboo, banana, and diatoms. It implied that some common and basic mechanisms, other than genes, might play important roles in the silicon uptake process for a wide range of plants [21]. In general, the weathering of continental crust provided the source materials for forming clays and soils, and the uptake of dissolved silicon by plants leads to more negative δ^30^Si values in plants [15,17,21,22,23]. Both De La Roach et al. and our previous work presented comprehensive studies on silicon isotope dynamic fractionation in solution and organism [16,78], where the process of silicon isotope dynamic fractionation is consistent with the Rayleigh fractionation model, the same as that in silica precipitation [16,102]. The Rayleigh fractionation model and relevant expressions are given below:

This fractionation factor can also be expressed as ε*; the per-mil enrichment factor between substrate and product may also be approximated as:(9)$$\varepsilon^{*}=\left(\delta^{30}{\mathrm{Si}}_{Dsi}-\delta^{30}{\mathrm{Si}}_{Dsio}\right)/\ln f$$
where *f* is the fraction of silicic acid remaining in solution, and δ^30^Si_DSi_ and δ^30^Si_DSio_ are the isotopic values of dissolved silicic acid at *f* and *f* =1, respectively.

In addition to the Rayleigh effect in the dynamic fractionation of silicon isotope during the biological uptake into various plants, other controlling factors should be considered to explain obvious differences in both δ^30^Si values and fractions of dissolved H_4_SiO_4_ from roots to stem and leaves [22,23]. As summarized in Table 1, the isotope fractionation of silicon during the biological uptake is much greater that in other abiotic processes. By comparing the Si isotope fractionation observed during the formation of sponge spicules and in silicification of diatom opal [19,20], it was proposed that the two groups evolved Si acquisition mechanisms with different impacts on silicon isotopes [25]: (i) Siliceous sponges originated in the Precambrian [134], with a relatively low affinity for Si [135], at a time when weathering of silicate rocks resulted in high oceanic Si(OH)_4_ [136]; (ii) diatoms evolved in the Jurassic with a higher affinity for Si, probably because biological utilization by other siliceous organisms resulted in lower oceanic Si(OH)_4_ [137]. The most recent study proposed that sponges have initiated seawater oxygenation by redistributing organic carbon oxidation through filtering suspended organic matter from seawater, and the resulting increase in dissolved oxygen levels potentially triggered the diversification of eumetazoans [138].

### 4.5. Silicon Cycling in Hydrosphere

#### 4.5.1. Global Spatial and Temporal Variability/Heterogeneity in Oceanic Si Cycle and δ^30^Si

As discussed above, both biotic and abiotic silicification enriches light silicon isotopes, lifting δ^30^Si and decreasing dissolvable Si (DSi) in the oceans. In a longer temporal scale, major geological (e.g., the banded iron formation (BIF)) and biological events (e.g., biosilicification) impacted on the global oceanic Si cycle. As shown in Figure 8a,b, the evolution of seawater δ^30^Si over geological time was derived from the recorded δ^30^Si in chert, radiolarian, and sponge spicules, diatoms, which shows a rising trend compared to the Archean ocean generally. Meanwhile, during much of the Archean, the banded iron formations (BIFs) that contain, on average, 30% iron and 40% or more Si were most deposited prior to the initial rise of atmospheric oxygen at ~2.45 Ga in marine basins in stratified water columns [61], which induced the dramatic decline in DSi. Since the Archean, the impact of the evolution of biosilicifying organisms on the DSi inventory of the oceans is hypothesized by Conley et al. [61], which concluded that biosilicification has driven variation in the global Si cycle, and bacterial silicon-related metabolism has been present in the oceans (Figure 8c,d).

In a spatial scale, both the DSi concentration and the silicon isotope values in a local ocean are also variable with the depth increasing in water columns, as shown in Figure 9. In marine environments, biogenic silica is primarily produced from planktonic organisms living in the surface ocean and a large fraction of biogenic silica is recycled via dissolution within the upper 100 m of the water column [89]. Silicic acid is an important nutrient in the surface oceans as it is required for the growth of diatoms, radiolarians, and silicoflagellates. In general, H_4_SiO_4_^0^ concentration of the surface is the lowest and increases from the surface to 1000 m (Figure 9a) [24,25,140,141]. In order to understand the controlling processes on the dissolution of biogenic silica and the buildup of silicic acids in marine sediments, the silica solubility of surficial bio-siliceous materials (e.g., Southern Ocean sediments, plankton, culture diatoms, and amorphous silica gel) as a function of temperature was plotted in Figure 9b. The equilibrium silica solubility increases with increases in environmental temperature, and is highly variable with variations in temperature, pressure, and differences in specific surface area and Al content of bio-siliceous fragments [91]. The silicon isotope composition of biogenic silica, opal, and dissolved silicic acid can reconstruct the global marine silicon cycle and consumption in deep ocean today and in the past [24,25,142,143]. Therefore, the relationships between H_4_SiO_4_^0^ and δ^30^Si in sponge spicules and diatoms are highly supportive of the qualitative use of diatom and sponge δ^30^Si as a proxy for the ambient DSi concentration in bottom water. Consequently, the silicon isotope compositions of marine waters are not homogeneous, where surface waters are enriched in ^30^Si relative to intermediate and deeper waters. For example, there is around 0.4‰ of difference in δ^30^Si observed between the measured deep waters from the central North Atlantic (+1.3 ± 0.2‰) [46] and central North Pacific (+0.9 ± 0.1‰) Oceans [46], suggesting that the isotopic composition of silicon in seawater is controlled not only by the isotopic signature of the inputs and outputs of silicon in ocean, but also by the interaction of the biologic cycling of silicon with the global thermohaline circulation of the sea.

#### 4.5.2. δ^30^Si Variation in Terrestrial Hydrosphere

In addition to marine system, the variation of δ^30^Si could help us evaluate the silicon budgets in terrestrial aqueous systems. After the pioneering works on the δ^30^Si of dissolved silicic acid in oceans, rivers, and estuaries [46], the δ^30^Si distributions in various fluids have been clarified, such as freshwater (i.e., −0.1 to +3.4‰) [18,26,46,49,50,51,112,118,141,148,149,150,151], groundwater (−1.43 to +0.43‰) [51], and seawater (i.e., +0.4 to +3.1‰) [13,19,46,47,140,152]. As discussed above, the preferential uptake of light Si isotopes into soil clays and organisms leads to predominately positive Si isotope signatures in continental surface waters. Further, the seasonal variation, the weathering degree, and mixing process also affect the silicon isotope compositions in surface water [150,151]. The precipitation of secondary clay minerals and the plants adsorption are accelerated under higher temperature, driving more positive shift of δ^30^Si in summer than winter, while higher weathering degree releases lighter silicon isotopes into surface waters from silicate minerals. Compared to surface waters, groundwater with lighter δ^30^Si values possibly is the result of secondary clay minerals and silcrete dissolution.

## 5. Geological Applications of Silicon Isotopes

### 5.1. Implication for Meteorites and Planetary Core Formation

#### 5.1.1. Origin of the Lunar Planetary Materials

The origin of the Moon is still debated. The similarity (or not) of the Moon to the Earth in terms of major element budgets and the heterogeneity degree of lunar mantle provide key information as regards the composition of the Moon-forming impact and/or the processes operating during lunar formation. There are a few reported high-precision silicon isotope compositions in lunar, such as the average δ^30^Si values of −0.31 ± 0.07‰ [50] and −0.30 ± 0.05‰ for lunar basalts, and a much lighter δ^30^Si value of −0.45 ± 0.05‰ for the lunar breccia 14304 [53]. More investigations reported the δ^30^Si values of −0.29± 0.06‰ for Low Ti basalt, 0.32± 0.09‰ for High Ti basalt, −0.29± 0.05‰ for lunar glass, and −0.27± 0.10‰ for highland rocks, and the data are in agreement with other isotope systems showing the majority of core formation happened early and before the giant impact [7].

The comparison of published δ^30^Si data relative to NBS–28 for samples from planetary bodies in the inner solar system concluded that the lunar mantle and BSE have similar δ^30^Si, and are significantly heavier than that in Mars, 4–Vestra and chondrite parent bodies (e.g., [7,50,55,153]), and the enstatite chondrites have the lightest Si isotope composition [8]. The study proposed two possible hypotheses to explain the apparent contradiction as: (i) At least part of the Earth’s building blocks originally had a heavy silicon isotope composition than that observed in chondrites; (ii) the mantle–core differentiation would generate obvious isotope fractionation, resulting in the observed heavy isotope composition of the bulk silicate Earth, if the Earth accreted only from chondritic materials. Further, the loss of light Si isotope during partial planetary vaporization in the aftermath of the Moon-forming giant impact could explain the similar heavy isotope composition of silicate portion of the Earth and the Moon [8].

#### 5.1.2. Formation of the Earth Core

It has long been proposed that the Earth’s core must contain a significant percentage of light elements to fit the density inferred from seismic studies. The light elements of hydrogen, carbon, oxygen, silicon, and sulfur are being considered [154]. According to the chondritic Mg/Si ratios in the terrestrial mantle, Si in particular is an important component in the core [155,156]. Because of the distinctive compositions of silicon in the core (i.e., 6 wt.%) and in the primitive mantle (i.e., 21 wt.%) [1] and the different partitioning behaviors of silicon with varying temperature, pressure, and oxygen fugacity (e.g., References [157,158,159,160]), silicon is a likely candidate for entering the core during metal–silicate differentiation of the Earth [7]. As the equilibrium isotope fractionation is driven by differences in bonding structure, it is presumed that differences in structure between silicate and metallic liquids would lead to silicon isotope fractionation [50]. Given that the silicon isotope fractionation between metal and silicate is the result of isotope equilibrium, the percentage of Si in the core could be calculated with the following equations (Equations (10)–(12)).
(10)$$\delta^{30}Si_{meteorite*}=f\delta^{30}Si_{BSE}+\left(1-f\right)\delta^{30}Si_{core}$$
(11)$$\Delta^{30}Si_{BSE-meteorite*}=\varepsilon \left(1-f\right)$$
(12)$$X=100\left(\frac{M_{BSE}}{M_{Core}}\right)\left[\frac{c}{1-\frac{\Delta^{30}Si_{BSE-meteorite*}}{\varepsilon }}-c\right]$$
where *f* is the fraction of Si in the silicate phase; *ε* is the fractionation factor of Si isotope between metal and silicate; M_BSE_ and M_Core_ are the mass fractions of BSE and the core; *c* is the measured Si fraction in BSE, which is identified as 0.212 [161].

The theoretical calculations provided the isotope fractionation factors of silicon (ε) is in the range of 0.4‰ to 2.0‰ in partitioning experiments [162,163]. Compared with the traditional index of superchondritic Mg/Si, the isotope fractionation of silicon would be dependent on temperature and pressure, providing constraints on mass balance and/or core formation conditions. A systematic investigation on silicon isotope compositions among 42 meteorite and terrestrial samples reported an average difference Δ^30^Si_BSE–meteorite_ of 0.15 ± 0.10‰, where the heavier δ^30^Si in BSE implied that Si partitioned into the metal phase during metal–silicate equilibrium at the time of core formation and the Δ^30^Si_BSE–meteorite_ value indicated that the silicon content is from at least 2.5 wt.% to 16.8 wt.% in the Earth’s core when fixing the temperature of core formation to the peridotite liquidus [7]. Relatively consistent Si–core concentrations of 13 wt.% [50], 6 wt.% [6], 1 wt.% [53,54], and 12 wt.% [8] are reported. As temperature and pressure are critical parameters for Si isotope fractionation during mantle–core differentiation, a temperature dependence of isotope fractionation of ∆^30^Si_silicate–metal_ = 7.64 × 10^6^/T^2^ in enstatite chondrites was measured by Ziegler et al. [56], which is in good agreement with independent experimental and theoretical determinations. Moreover, they also calculated the wt. % of Si in the core and Δ^30^Si_BSE–chondrite_ as the functions of ∆IW (i.e., the fugacity of oxygen defined by the reaction: Fe + ½O_2_ = FeO) and P(T) along magma ocean adiabat using the thermodynamics of Si solubility in metal established experimentally [56,164]. A comprehensive investigation on the silicon isotope variation in enstatite meteorites revealed that the meteorites are the lightest macroscale solar system objects in Si isotopes (e.g., −0.77 ± 0.08‰ in EH chondrites, −0.59 ± 0.09‰ for EL chondrites, and −0.60 ± 0.11‰ in aubrites), and the similarity of δ^30^Si in metal free component of EH, EL, and carbonaceous/ordinary chondrite indicated that the Si isotope variation in the nebular gas did not induce the light Si isotope enrichment in enstatite chondrites [57]. The driving force for the light Si isotope enrichment in enstatite chondrites was explored as a result of refractory lithophile element fractionation according to the variations in Mg/Si and Ai/Si ratios in the order of carbonaceous–ordinary–enstatite in chondrite rocks [57]. Recently, the equilibrium isotope fractionation during the high-temperature partitioning of stable isotopes of rock-forming elements (e.g., Mg, Si, Fe, H, O) has been reviewed by Young et al. [165]. The limiting parameters of using silicon isotopes to constrain the conditions for core formation are the uncertainty in Δ^30^Si_BSE–BE_ because disagreement about the typical ^30^Si/^28^Si of chondrites and the inevitable uncertainty as to whether bulk Earth is truly chondritic in silicon isotope rather than the fractionation factor.

### 5.2. Implication for Core Deposits Formation and Hydrothermal Fluids Activities

#### 5.2.1. BIF Deposits

Banded iron formations (BIFs) are chemical marine sediments that formed periodically throughout the Precambrian (3.8–0.5 Ga) and are usually characterized by alternating Fe- and Si-rich layers. The peak in BIFs formation between 2.5 and 2.3 Ga appears to correlate with major changes in the Earth’s history such as the rise of atmospheric oxygen and the change from anoxic to oxic conditions in the ocean [166]. It has been found that chert within BIFs exhibits a largely negative silicon isotope signature ranging from about −2.5‰ to −0.5‰ in δ^30^Si which has been interpreted as a hydrothermal signal [10,32,167,168]. Positive δ^30^Si values in Precambrian chert may reflect elevated temperature of the seawater, the influence of a continental source, or precipitation from isotopically heavy seawater [70,110,169,170]. In Precambrian chert deposits, silicon isotopic variations at several localities correspond to Al_2_O_3_ contents and rare earth element patterns [101,105,110,170], indicating that (near-) primary signatures are often preserved, even though diagenesis may have affected primary signatures. The silicon isotope compositions of S-cherts show a limited range in δ^30^Si values from +0.1‰ to +1.1‰, linking to the silicification of volcanic sedimentary precursor rocks [105,110,169]. In contrast, larger kinetic isotope effect inferred for this pathway of chert formation is consistent with the strongly negative δ^30^Si-values reported for C-chert, which are often considerably lower than the value of −0.3‰ in modern deep-sea hydrothermal fluids, assuming that this was similar in Precambrian times [46]. The C-chert with low Al_2_O_3_ content showed obvious ^30^Si depleted compositions, e.g., down to −3.7‰ for Eoarchean banded iron formations from Isua, Greenland [108], −2.4‰ for Paleoarchean chert from the Pilbara Craton, Western Australia [105,171], −2.6‰ for Paleoarchean chert from South Africa and Zimbabwe [101,172,173], and −4.3‰ for Proterozoic banded iron formations from Western Australia [169]. Therefore, the origin of these cherts has been linked to chemical precipitation of silica from mixtures of hydrothermal fluids and seawater [37].

A co-variation of Fe and Si isotope compositions was observed in a magnetite–carbonate–chert BIFs from the Archean Old Wanderer Formation in the Shurugwi Greenstone Belt, which was interpreted as largely primary signatures [173]. The significant variations of δ^30^Si from −1.0‰ to −2.6‰ in bulk layers suggested rapid precipitation of the silicate phases from hydrothermal-rich waters, and the changes of Fe and Si isotope signatures directly reflect the upwelling dynamics of hydrothermal-rich water, which govern the rates of Fe and Si precipitation and the development of layering [173]. The combination of rare-earth-elements and yttrium patterns and higher silicon isotope compositions (up to ca. +0.9‰) confirmed the marine origin of the chert across the ~3.42 Ga in Buck Reef Chert (BRC) in the Barberton Greenstone Belt (South Africa), and the black chert bands with consistently higher δ^30^Si than coexisting translucent counterparts at the same stratigraphic level was interpreted as a primary feature acquired during deposition upon interaction between submarine discharging hydrothermal water and a stratified water body [103].

The metallogenic models of two typical chert deposits (i.e., S-chert, C-chert) are schematically shown in Figure 10, in which the insert shows the distribution pattern of δ^30^Si vs. Al_2_O_3_%. Clearly, the C-chert with negative δ^30^Si and lower Al_2_O_3_ contents (less than 0.2%) indicates the non–equilibrium condition during rapid chemical precipitation caused by oversaturated fluid mixtures at relatively low ambient seawater temperature [109] during upwelling hydrothermal fluids with higher temperature and acidic pH. Both the dike S-chert and stratiform S-hert show positive δ^30^Si characters and the correlation of δ^30^Si and Al_2_O_3_ content implies that the relative contribution from isotope effect within silicification and precursor rocks with negative δ^30^Si values (−0.29 ± 0.08‰) [3,115]. Those understanding would greatly enhance the viability of silicon isotopes in cherts for reconstructing the evolution of ancient marine basins.

#### 5.2.2. Hydrothermal Polymetallic Core Deposits

Silicon isotopic measurements used in combination with other geochemical indices (e.g δ^18^O and δ^11^B) can be used to better constrain the genesis of ore deposits. The origin of polymetallic ore deposits (e.g., the Dachang Sn-polymetallic ore deposit in China) and the Sullivan Pb–Zn ore deposit (British Columbia, Canada) have been well clarified. The Dachang Sn-polymetallic ore deposit (Guangxi, China) is one of the largest Sn deposits in the world, and the second largest Sn producer in China (more than 106 tons of tin) [34]. There are various models proposed in order to explore the ore genesis of the deposit, and the major debates are focusing on whether the stratiform Sn–Pb–Zn orebodies are syn-sedimentary in origin or products of the Yanshannian magmatic-hydrothermal event (~100 Ma) [34]. A systematic survey on silicon and oxygen isotopes among siliceous rocks had been carried out by Ding et al. [10], and the distribution of δ^30^Si–δ^18^O values constrained the two individual origins of siliceous rocks: (i) Shallow radiolarian siliceous rocks without ore deposits. It is widely distributed with larger thickness and exhibits higher δ^30^Si (+0.2 ~ +0.8‰) and wider variation of δ^18^O values (+12.0 to +21.4‰). There is no direct relationship between the radiolaria siliceous rocks and mineralization, but the distribution along the Devonian deepwater rift trough might indicate the possible submarine exhalative environment. The existence of large amount of radiolaria in these rocks further approved the biochemical deposition genesis. (ii) The thin-laminated siliceous and K-feldspar rocks associated with cassiterite-sulfide ores. The negative δ^30^Si (i.e., −0.1 to −0.6‰) and uniform δ^18^O values (i.e., +13.2 to +16.0‰) mostly favored a submarine hydrothermal origin rather than the Yanshannian magmatic–hydrothermal origin [10,33,174]. The petrology, chemistry, Si–O isotope distribution of siliceous rocks, and Sr–Nd isotope systematics of tourmalines in the Dachang Sn-polymetallic ore deposit do not support a skarn or replacement-type origin [175] but are more compatible with a submarine exhalative-hydrothermal origin [176].

The Sullivan Pb–Zn ore deposit (British Columbia, Canada) is a classic example of stratiform sediment-hosted Pb–Zn deposit, and silicate minerals associated with the formation and post-depositional evolution of the orebody are widespread throughout the deposit [33]. The clastic sedimentary rocks of the lower Aldridge formation that host the Pb–Zn orebody have the δ^30^Si values of 0 to +0.1‰, similar to that obtained from other clastic sediments [10,31,32]. The silicon isotope compositions of tourmaline supported that tourmalinites from deep in the feeder zone were formed by the replacement of clastic sediments, and stratiform and garnet–rich tourmalinites in the immediate footwall of massive sulfide orebody were formed mainly by hydrothermal exhalative processes [33], which is consistent with the genetic model proposed by Slack [177].

#### 5.2.3. Clay Minerals Deposits

The source and transformations of silicon in global biogeochemical cycles are complex, particularly in terrestrial settings where it can occur in numerous forms: Primary (unweathered) minerals, secondary crystalline minerals and non–crystalline forms, biogenic silica (phytoliths), and aqueous phases (Si(OH)_4_). A frontier study was to track the source of secondary clay minerals (e.g., kaolinite) using silicon isotope composition [10]. A comparative δ^30^Si survey on the secondary minerals from Southeast China, USA, and Italy indicated that the δ^30^Si in clay minerals varied in a wide range from −1.9 to +0.1‰ (i.e., 2.0‰ in total). Clay minerals from different sources can be distinguished by the δ^30^Si characters (Figure 11). Kaolinite in hydrothermal alteration genesis has a narrow δ^30^Si value from −0.1‰ to +0.1‰, consistent with that in granite, indicating that no significant silicon isotope fractionation occurs during the formation of kaolinite. A wide variation of δ^30^Si values (−1.9 to −1.0‰) in kaolinite in weathered origin is observed, and the δ^30^Si values shift negatively with increasing of weathering degree, which reflect a significant isotope fractionation during weathering, as discussed in Section 4.3 above. The sedimentary kaolinite with the δ^30^Si values of −1.2 to −0.1‰ are between hydrothermal kaolinite and weathering kaolinite, perhaps showing mixed origins. Therefore, it is possible to constrain the origins of clay minerals in accordance using the δ^30^Si characters [10].

## 6. Conclusions

In this work, the fundamental advances in silicon isotope geochemistry investigations have been reviewed, including the silicon isotope fractionations linked to silicon complexations/coordination and thermodynamic conditions in various geological processes (i.e., high-temperature processes, low-temperature precipitation, chemical weathering of crustal surface silicate rocks, biological uptake), and the geological implications for meteorites and planetary core formation, ore deposits formation and hydrothermal fluids activities, weathering rate, and silicon cycling, etc. On the basis of the distribution of silicon isotope compositions and the isotope fractionations associated with silicon transportation and cycling among different geological processes, the following issues could be concluded:

The silicon isotope fractionations in various geological processes are distinctive: (i) In the high-temperature process, the equilibrium silicon isotope fractionation during the mantle–core differentiation results in the observed heavy isotope composition of the bulk silicate Earth (BSE) if the Earth accreted only from chondritic materials; in low-temperature precipitation, it is strongly dependent on the degree of (metastable) equilibrium in the silica–water system (e.g., distribution of speciation of H3 and H4, solubility, temperature, pH, etc.) and the presence of carrier phase (e.g., Al–hydroxide, Fe–hydroxide); (ii) the equilibrium fractionation of silicon isotopes among silicate minerals are sensitive to the Si–O bond length, Si coordination numbers (CN), the polymerization degrees of silicate units, and the electronegativity of cations in minerals. (iii) In chemical weathering processes, the isotope fractionation is controlled by the associated dynamic fractionation in precipitation during the weathering of silicate minerals, the selective adsorption of H_4_SiO_4_^0^ on soil Fe–oxides/Al–oxides, and the degree of soil desilication; (iv) in the biological uptake processes, biological fractionations of silicon isotopes are very poorly understood, largely because the complicated biochemical pathways, likely involving several enzymatic processes and active transport proteins.

With further understanding on the driving mechanisms for silicon isotope fractionations behind typical geological processes, the silicon isotope geochemistry provides powerful geochemical constraints for tracing bio-physico-chemical processes in terrestrial environments, weathering processes, mineral deposits formation, hydrothermal fluids activities, and meteorite and planetary evolutions, etc. (i) The comparison on δ^30^Si values in various lunar materials (e.g., lunar mantle, lunar basalts, lunar breccia 14304, lunar glass, highland rocks, etc.,) and BSE approved the majority of lunar materials formed early and before the giant impact. (ii) The equilibrium silicon isotope fractionation during the mantle–core differentiation provided the potential to evaluate the Si-core concentration and to explore the driving force for the light Si isotope enrichment in enstatite chondrites. (iii) The combination of δ^30^Si and other geochemical indices would well constrain the genesis of ore deposits and the hydrothermal fluids activities. For instance, the metallogenic models of two typical chert deposits (i.e., S-chert, C-chert) could be identified in accordance with the distribution pattern of δ^30^Si vs. Al_2_O_3_%. The origin of polymetallic ore deposits (e.g., the Dachang Sn-polymetallic ore deposit in South China) has been well clarified with Si–O isotopes in siliceous rocks, Sr–Nd isotope systematics of tourmalines, and chondrite–normalized rare earth elements (REE) patterns. The δ^30^Si has the potential to be a useful weathering proxy, tracking the origins of clay minerals and desilication degree of soils.

In the future work, it is urgent to provide rigorous experimental and/or theoretical studies to calibrate critical effects on isotopic fractionation during melting, core formation, and differentiation, and to better understand the mechanism of dynamic isotope fractionation of silicon during silicon cycling from lithosphere to biosphere and transportation in mineral digenesis.

## Figures and Tables

**Figure 1 molecules-24-01415-f001:**
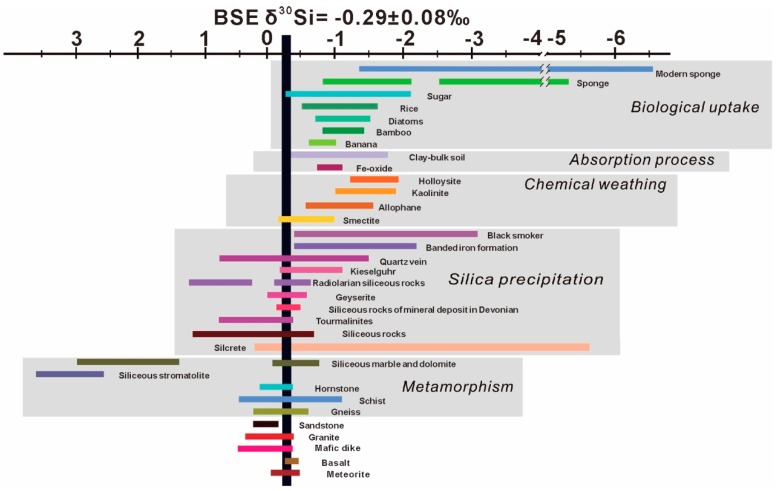
Silicon isotope variations as function of host rocks, organisms, and corresponding geological processes in nature (modified with permission from Wang et al. [78], Langmuir, published by ACS, 2016).

**Figure 2 molecules-24-01415-f002:**
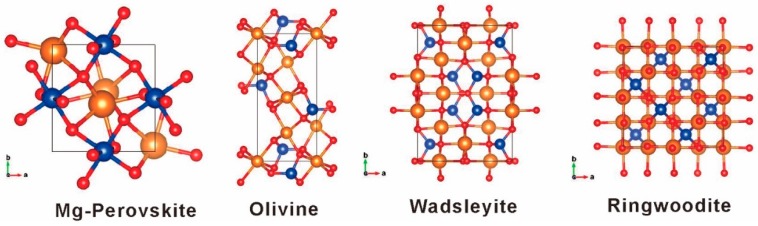
Crystal structures of perovskite (Mg^VI^SiO_3_), olivine (Mg_2_^IV^SiO_4_), wadsleyite (Mg_2_^IV^SiO_4_), and ringwoodite (Mg_2_^IV^SiO_4_). Magnesium, oxygen, and silicon atoms are represented by orange, red, and blue color individually.

**Figure 3 molecules-24-01415-f003:**
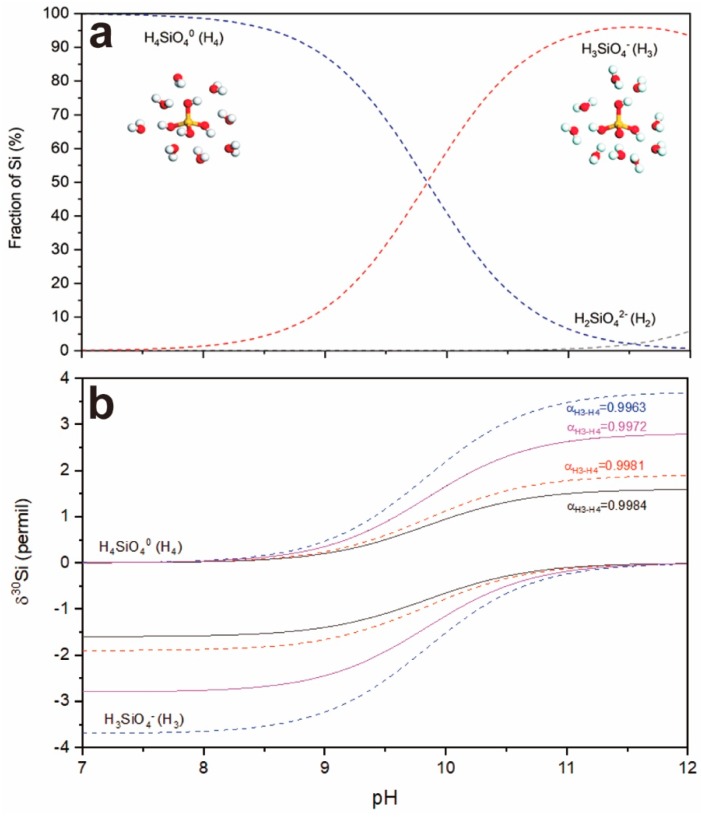
(**a**) Distribution of H4, H3, and H2 species in aqueous solution as a function of pH according to the dissociation constants of pK_a1_ = 9.84 and pK_a2_ = 13.2 at 298 K and 0 M ionic strength condition Fujii et al. [86]. Inserted are structures of hydrated silicic acid H_4_SiO_4_^0^ (H4) and its hydrated dominant-associated base H_3_SiO_4_^−^ (H3) in aqueous solution. Silicon, oxygen, and hydrogen atoms are represented by yellow, red, and white color. (**b**) δ^30^Si of H4 and H3 species relative to the bulk solution (δ^30^Si = 0) with α_H3-H4_ values of 0.9963, 0.9972, 0.9981, and 0.9984 in the solution pH range of 7–12. The α_H3-H4_ values used were referred to the previous study by Fujii et al. [86].

**Figure 4 molecules-24-01415-f004:**
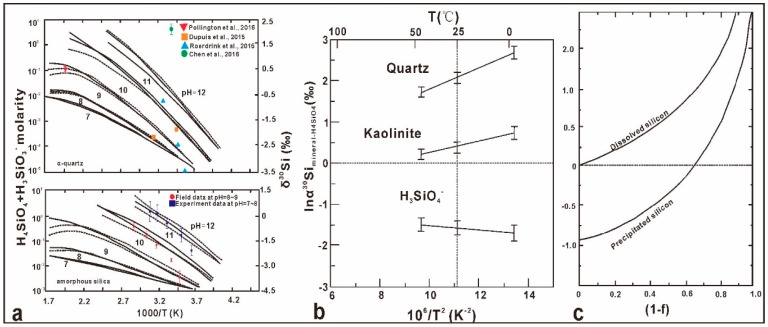
(**a**) Solubility of α-quartz and amorphous silica as a function of temperature at various pH. Dotted line: γ = 1.0. Dashed line: γ in Fleming et al. 1982 (data compiled from References [29,37,87,92,103,106,107]); (**b**) calculated equilibrium fractionation factors of silicon factors between minerals (quartz and kaolinite) and dissolved silicon speciation of H_4_SiO_4_^0^, and between H_4_SiO_4_^0^ and H_3_SiO_4_^−^(modified with permission from Dupuis et al., Chem. Geol. published by Elsevier, 2015); (**c**) kinetic fractionation of silicon isotopes in silica precipitation process from solution at pH 7 in room temperature with the α_precipitated-dissolved_ factor of 0.9990 given by Li et al. [102]).

**Figure 5 molecules-24-01415-f005:**
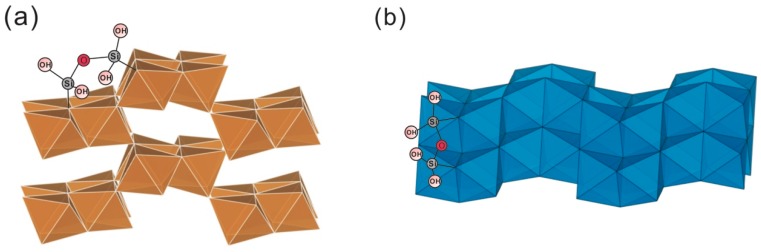
Schematic illustration of bonding structures of polymerized silicic acid with goethite (**a**) and ferrihydrite (**b**) minerals. Octahedra represent the atomic shells in iron oxy-hydroxide minerals.

**Figure 6 molecules-24-01415-f006:**
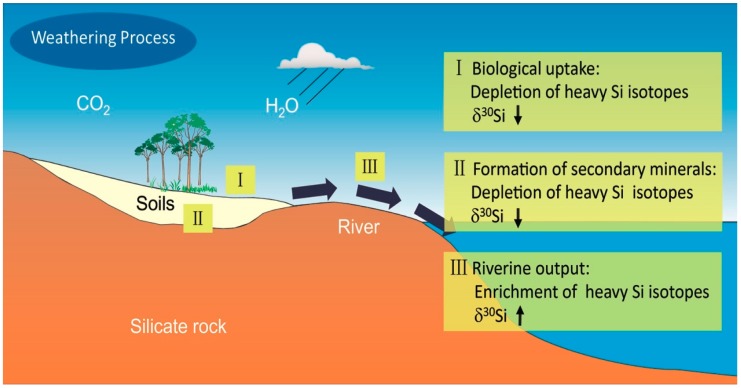
Geological silicon cycling and associated isotope fractionations during the crustal surface weathering processes.

**Figure 7 molecules-24-01415-f007:**
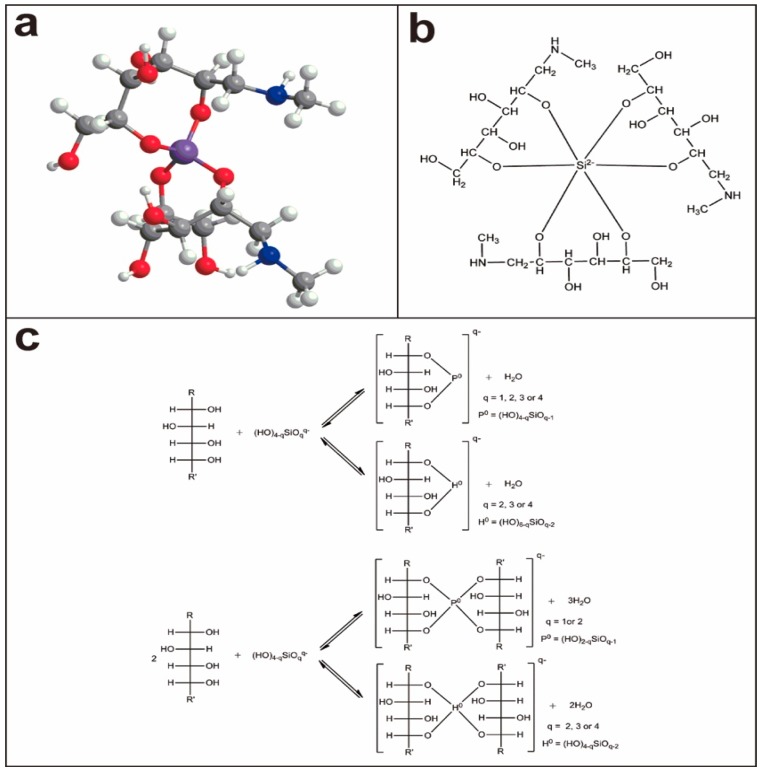
Molecular structures of organo-silicon complexes. Silicon is tetracoordinated (**a**) and hexacoordianted (**b**) with sugar; and in stable hypervalent (**c**) with straight-chain polyhydroxy compounds (data compiled from [78,127,128,129,130,131,132]).

**Figure 8 molecules-24-01415-f008:**
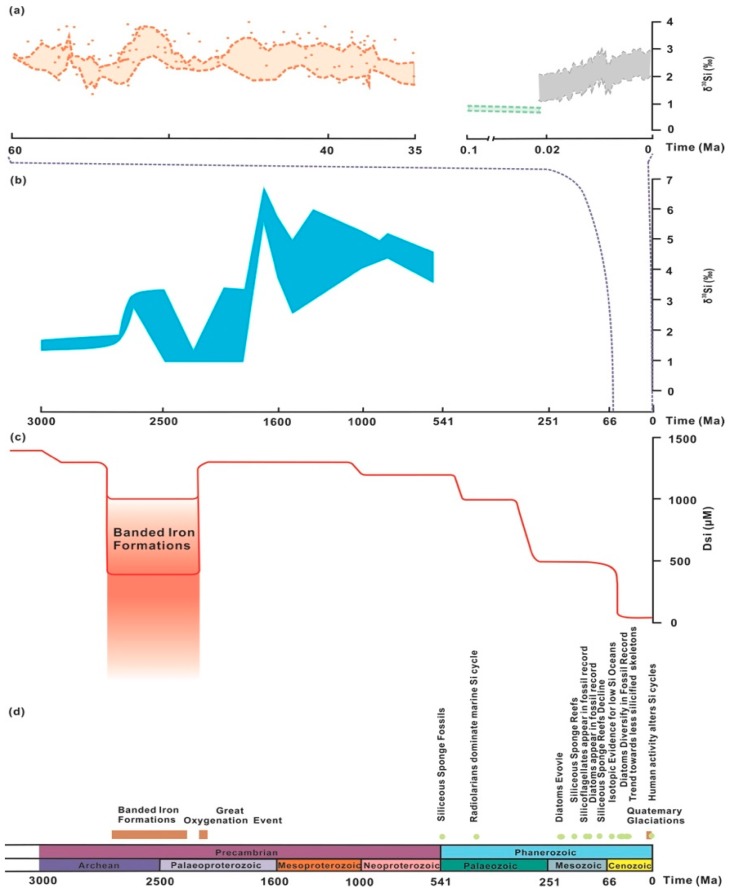
(**a**) Variations of seawater δ^30^Si from 60 Ma to the present. Variations in the period of 65–30 Ma is derived by correcting δ^30^Si in radiolarian and sponge spicules, as reported by Fontorbe et al. [144], using the fractionation factor α_sponge–solution_ of 0.9979. Seawater δ^30^Si in the periods of 100–20 ka is cited from De La Rocha and Bickle [145] and the up-limit and the down-limit of seawater δ^30^Si in 20–0 ka are derived based on the δ^30^Si measured from core samples by Horn et al. (2011) [146] with the fractionation factor α_diatom–solution_ of 0.9985 and 0.9993, respectively; (**b**) evolution of seawater δ^30^Si recorded from Precambrian chert from 3000 Ma to 500 Ma (data compiled from Marin-Carbonnea et al. (2014) [147]; (**c**) evolution of dissolvable Si (DSi) content in ocean responding to dominant geological event (e.g., the banded iron formation (BIF)) and biological events (e.g., biosilicification) from the early Precambrian to the present (modified with permission from Conley et al. Fronters Marine Sci. published by Frontiersin 2017);.(**d**) Significant geological and biological events that affect the global oceanic Si cycle (modified with permission from Conley et al. Frontiers Marine Sci. published by Frontiersin 2017).

**Figure 9 molecules-24-01415-f009:**
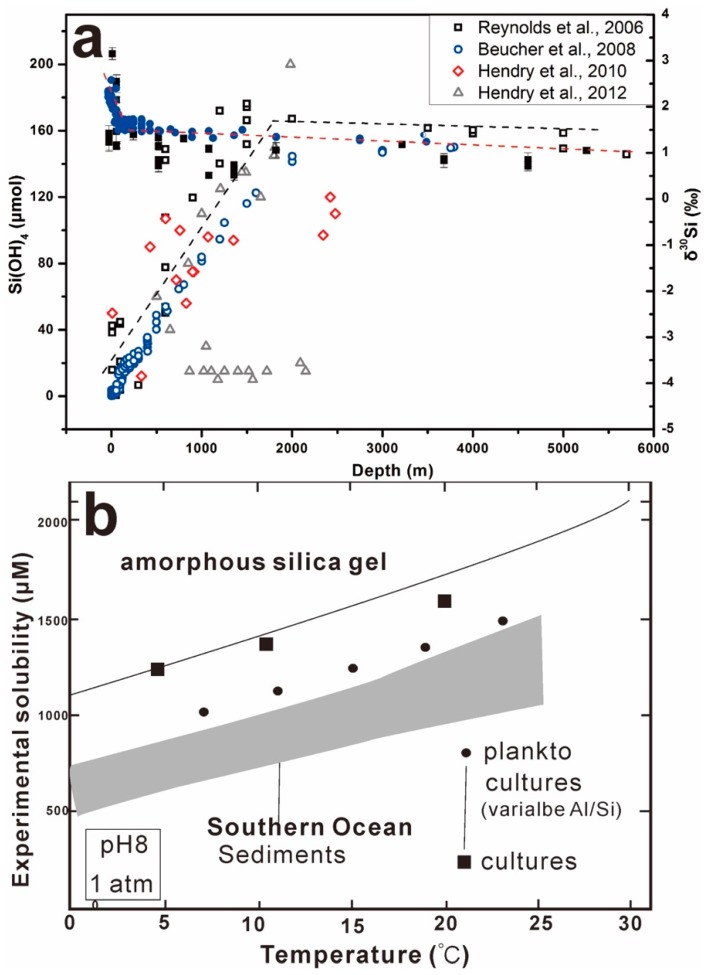
(**a**) Variations in Si(OH)_4_ concentration and the silicon isotope values in seawater with the depth increasing in water columns [24,25,140,141], open points represent the Si(OH)_4_ concentration and half open points represent the silicon isotope value; (**b**) calculated bulk solid solubility of biogenic silica plotted against temperature using the solid-solution interfacial free energy of amorphous silica ( modified with permission from Dixit et al., Mar. Chem. published by Elsevier 2001).

**Figure 10 molecules-24-01415-f010:**
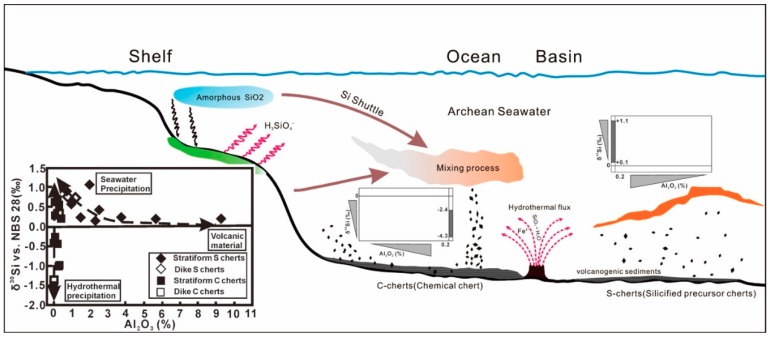
Schematic diagram for the metallogenic models of two typical BIF deposits (C-chert, S-chert). The insert shows the distribution pattern of δ^30^Si vs. Al_2_O_3_% among stratiform and dike C- and S-chert. Data compiled from van den Boorn et al. [109] and Swedlund and Webster [115].

**Figure 11 molecules-24-01415-f011:**
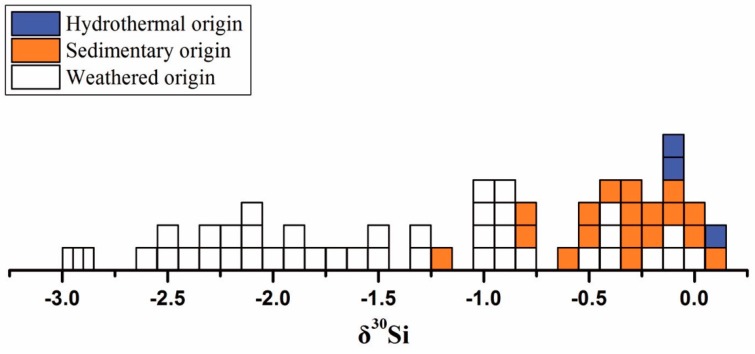
Distributions of δ^30^Si among various types of clay minerals (data cited from References [9,10,12,26,89,118].

**Table 1 molecules-24-01415-t001:** Silicon isotope fractionations between various biotic and abiotic materials and solution.

Biotic and Abiotic Material	Δ^30^Si (‰)	α_solid-solution_	References
**Biological Uptake**			
Sugar	−0.26 to −2.09	0.9980	[98]
Rice	−0.5 to −1.6	0.9995–0.9984	[21]
Bamboo	−0.8 to −1.4	0.9992–0.9986	[22]
Banana	−0.6 to −1.0	0.9994–0.9990	[23]
Diatoms	−0.7 to −1.5	0.9993–0.9985	[16,19,20]
	−1.3 to −0.9	0.9987–0.9991	[16]
Sponge	−0.8 to −2.1		[19,20]
	−2.5 to −5.3	0.9960–0.9993	[24]
	−1.32 to −6.52	0.9934–0.9986	[25]
**Adsorption Process**			
Fe–oxide	−0.73 to −1.09	0.9992–0.9995	[112]
Gibbsite (γ–Al(OH)_3_)		0.9970–0.9982	[11]
Clay–bulk soil	−0.29 to −1.74	0.9993–0.9997	[12,13]
**Precipitation**			
Abiotic silica	−1.3 to −3.8	0.9990–0.9996	[102]
Biogenic silica	−1.93 to −1.33		[139]
Euhedral megaquartz	−1.8 to −2.1		[105]
**Chemical Weathering**			
Smectite	−0.16 to −0.52		[51]
Kaolinite	−2.2		[26,118]
	−1.9 to +0.1		[10]

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
