# Peer review of "Silicon Isotope Geochemistry: Fractionation Linked to Silicon Complexations and Its Geological Applications"

_molecules, 2019, doi:10.3390/molecules24071415_

Round 1

Reviewer 1 Report

The aim of this paper is to review silicon isotope fractionation in natural systems, with specific reference to complexation and coordination. There have been a number of reviews in the last five years on silicon isotope fractionation from the point of view of observations of high-temperature processes (Savage et al., 2014), terrestrial cycling (Frings et al., 2016), marine cycling and palaeoclimate (Hendry & Brzezinski, 2014), and – indeed – more wide ranging reviews covering a number of aspects of the field (Poitrasson et al., 2017; Sutton et al., 2018). However, to the best of my knowledge there’s not been a recent review paper that specifically compares and contrasts observations and theoretical calculations/computational geochemistry. As such, I could potentially see a “place” for a paper such as this one in the literature. However, I think that there is currently too much information not strictly relevant to the main point of the paper, making it too similar to recent review papers. For example, there is no need to have a full description of the history of analytical methods (section 2), or go into full details of the global silicon cycle and reservoirs (section 3). The paper would be greatly improved by a complete restructuring and a rewrite, to be more concise and to have all the information related directly back to the key message: that complexation and co-ordination are key factors in silicon isotopic systematics. The authors should expand upon more of the experimental and computational findings (e.g. section 4), perhaps also discussion some of the limitations of the approaches and more specific future avenues of research, and make more of comparing to field observations. Also, there are a considerable number of poorly phrased (and very long) sentences, typos, and other grammatical errors that need fixing before this manuscript it accepted for publication. As such, I’d recommend an ab initio submission with a major overhaul, keeping in mind the central tenet of the paper: complexation and co-ordination are inherently relevant to isotopic studies.

Most of the information in the manuscript appears to be sound (although there are parts that are difficult to follow because of the confusing use of track changes). The one exception to this is the explanation of biological fractionation of silicon isotopes. It should be noted that the role of complexation in biological fractionation of silicon isotopes is very poorly understood (largely because the complicated biochemical pathways themselves, likely involving several enzymatic processes and active transport proteins, are not well constrained in any siliceous organism). However, what is clear is that isotopic behaviour during biosilicification is not always explained by a Rayleigh-type fractionation model (e.g. sponge spicule formation). Furthermore, the distribution of silicon and silicon isotopic variations in the oceans is not clearly explained. Figure 9 is rather confusing, and I’m not entirely sure I understand which data have been plotted here. The caption suggests that these are seawater silicon isotope data, but the isotopic range suggests that they are biogenic silica data as well (?). It’s very unclear as to what the underlying message of this figure might be.

Some examples of poor use of English:

I will not be able to provide a rigorous review of the grammatical errors in the manuscript. However, I will illustrate a few key points below:

1)    Very long sentences should be divided, or cut down. e.g. the first sentence of the abstract.

2)    Some statements do not follow logically e.g. line 61 – it does not follow that because there is a large budget of Si in the mantle reservoir that high-temperature fractionation is smaller (and smaller than what?). Also, line 160 – the use of HF does not result in the listed polyatomic interferences: those interferences are present when using any wet chemistry approach (e.g. alkaline digestion, alkaline fusion etc.).

3)    Be careful with sentence structure and use of (small) words e.g. line 209 – “silicon is the defining element of silicate reservoirs in Earth” should read “on Earth”. (As an additional aside, I’m also not entirely convinced I understand the point being made in this sentence). E.g. line 274 “with referring to previous study by…” should read “with reference to the previous study by…. E.g. line 359 – “always priority” should read “always the priority” (or probably better written as “preferentially taken up”), and so on.

4)    Be careful with labelling and referencing to figures. It’s incredibly difficult to follow in the PDF I have because of the track changes, but I think some of the figures are referenced incorrectly in the text e.g. line 263 refers to figure 4b, but I think this should be figure 3?

5)    There are a number of spelling mistakes that need fixing e.g. line 404 (polyermized) and line 408 (negatively charged) [sic].

6)    Be careful with author names e.g. line 406 (should read De La Rocha) and in the caption for figure 8 (should read De La Rocha and Bickle).

7)    Please ‘accept’ all track changes before resubmitting (and be clear as to the links with the supplementary information).

Author Response

The aim of this paper is to review silicon isotope fractionation in natural systems, with specific reference to complexation and coordination. There have been a number of reviews in the last five years on silicon isotope fractionation from the point of view of observations of high-temperature processes (Savage et al., 2014), terrestrial cycling (Frings et al., 2016), marine cycling and palaeoclimate (Hendry & Brzezinski, 2014), and – indeed – more wide ranging reviews covering a number of aspects of the field (Poitrasson et al., 2017; Sutton et al., 2018). However, to the best of my knowledge there’s not been a recent review paper that specifically compares and contrasts observations and theoretical calculations/computational geochemistry. As such, I could potentially see a “place” for a paper such as this one in the literature. However, I think that there is currently too much information not strictly relevant to the main point of the paper, making it too similar to recent review papers. For example, there is no need to have a full description of the history of analytical methods (section 2), or go into full details of the global silicon cycle and reservoirs (section 3). The paper would be greatly improved by a complete restructuring and a rewrite, to be more concise and to have all the information related directly back to the key message: that complexation and co-ordination are key factors in silicon isotopic systematics. The authors should expand upon more of the experimental and computational findings (e.g. section 4), perhaps also discussion some of the limitations of the approaches and more specific future avenues of research, and make more of comparing to field observations

  According to the valuable comment, the theoretical understanding on equilibrium silicon isotope fractionations among various silicate minerals have been enhanced (Line 242-261). As stressed in the Introduction, the general advances in silicon isotope geochemistry in the early period from 1950s to 1990s have been well demonstrated by Ding et al. The recent understandings on silicon isotope geochemistry have been systematically reviewed by individual scientists, such as the silicon isotope fractionation behavior during magmatic differentiation by Savage et al., the elemental and isotopic abundances in extraterrestrial and terrestrial reservoirs processes by Poitrasson et al., the continental Si cycle and its impact on the ocean Si isotope budget by Frings et al., the influence of the evolution of bio-silicifying organisms on oceanic dissolved Si inventory by Conley et al., as well as the latest understanding of Si cycle in marine, atmospheric, freshwater and terrestrial systems by Sutton et al. On the basis of the previous contribution, this review mainly focused on the fundamental advances in analytical approaches, the silicon isotope fractionation linked to silicon complexation/coordination and thermodynamic conditions in various geological processes (e.g. silicate minerals with variable structures and chemical compositions, silica precipitation and diagenesis, chemical weathering of crustal surface silicate rocks, biological uptake, global oceanic Si cycle), and their geological implications (Line 81-121).

  As a review paper, it is hard to discuss the progresses in the fields without the fundamentals of silicon isotope variations in major reservoirs. In addition, the advances and/or extensions of silicon isotope geochemistry studies fully depended on the improvement of analytical approaches. Therefore, the relevant sections (2. Analytical techniques, and 3. Silicon isotope variations in major reservoirs and geological processes) were kept in the revised version. The most parts of the review fully stressed on the key message-silicon isotope fractionation linked to silicon complexation/coordination and thermodynamic conditions in various geological processes (e.g. silicate minerals with variable structures and chemical compositions, silica precipitation and diagenesis, chemical weathering of crustal surface silicate rocks, biological uptake, global oceanic Si cycle), including 8 figures and 1 table (Line 214-600).

  In Section 4.5, the global Si cycle and δ30Si variation over large geological time scale and local scale have been addressed. The variations in both the dissoluble Si concentration and the silicon isotope values in a local ocean with the depth in water columns was illustrated in Figure 9, which aids to interpret the driving forces for such spatial variations in modern ocean (Line 592-608).

  Thanks for the rigorous review. The graph has been replotted, in which Figure 9a and 9b were laid out vertically, and the data points of δ30Si from biogenic silica have been removed in Figure 9a in the revised version.   

. Also, there are a considerable number of poorly phrased (and very long) sentences, typos, and other grammatical errors that need fixing before this manuscript it accepted for publication. As such, I’d recommend an ab initio submission with a major overhaul, keeping in mind the central tenet of the paper: complexation and co-ordination are inherently relevant to isotopic studies.

  According to the comment, an exhaustive grammatical editing through the whole text has been made by Dr. Simon V. Hohl. All issues related to the grammatical errors raised by the reviewers have been carefully corrected in the revised version.

Most of the information in the manuscript appears to be sound (although there are parts that are difficult to follow because of the confusing use of track changes). The one exception to this is the explanation of biological fractionation of silicon isotopes. It should be noted that the role of complexation in biological fractionation of silicon isotopes is very poorly understood (largely because the complicated biochemical pathways themselves, likely involving several enzymatic processes and active transport proteins, are not well constrained in any siliceous organism). However, what is clear is that isotopic behaviour during biosilicification is not always explained by a Rayleigh-type fractionation model (e.g. sponge spicule formation). Furthermore, the distribution of silicon and silicon isotopic variations in the oceans is not clearly explained. Figure 9 is rather confusing, and I’m not entirely sure I understand which data have been plotted here. The caption suggests that these are seawater silicon isotope data, but the isotopic range suggests that they are biogenic silica data as well (?). It’s very unclear as to what the underlying message of this figure might be.

   With the suggestive comment, the relevant interpretations on silicon isotope fractionation during biological uptake have been revised. Two individual parts have been complemented in this section: (i) “The role of complexation in biological fractionations of silicon isotopes is very poorly understood (largely because the complicated biochemical pathways, likely involving several enzymatic processes and active transport proteins, are not well constrained in any siliceous organism. Besides, it was found that silicon isotope fractionations between plants and growth solutions were basically similar in both direction and extent for rice, bamboo, banana and diatoms. It implied that some common and basic mechanisms, other than genes, might play important role in the silicon uptake process for a wide range of plants [21]” (Line 535-541); (ii) “In addition to the Rayleigh effect in the dynamic fractionation of silicon isotope during the biological uptake into various plants, other controlling factors should be considered to explain different δ30Si values and fractions of dissolved H4SiO4 from roots to stem and leaves [22, 23]” (Line 553-555).

Some examples of poor use of English:

I will not be able to provide a rigorous review of the grammatical errors in the manuscript. However, I will illustrate a few key points below:

1)      Very long sentences should be divided, or cut down. e.g. the first sentence of the abstract.

  As the reviewer suggested, the first sentence has been divided as three short ones “Firstly, the continuous modifications in analytical approaches and the silicon isotope variations in major reservoirs and geological processes have been briefly introduced. Secondly, the silicon isotope fractionation linked to silicon complexation/coordination and thermodynamic conditions have been extensively stressed, including silicate minerals with variable structures and chemical compositions, silica precipitation and diagenesis, chemical weathering of crustal surface silicate rocks, biological uptake, global oceanic Si cycle etc. Finally, the relevant geological implications for meteorites and planetary core formation, ore deposits and hydrothermal fluids activities and silicon cycling in hydrosphere have been summarized” (Line 16-23).

  In addition to this one, all long sentences in the whole text have been revised accordingly.

2)      Some statements do not follow logically e.g. line 61 – it does not follow that because there is a large budget of Si in the mantle reservoir that high-temperature fractionation is smaller (and smaller than what?). Also, line 160 – the use of HF does not result in the listed polyatomic interferences: those interferences are present when using any wet chemistry approach (e.g. alkaline digestion, alkaline fusion etc.).

    With referring to the comment, this part has been revised as “Because of the large budget of the mantle Si reservoir, and the long history of convective mixing [3], it has been widely recognized that the thermodynamic isotope fractionation of silicon associated with high-temperature processes is smaller than low-temperature” (Line 65-67).

  According to the comment, the previous wrong statement has been corrected as “The digestion of silicate materials using hydrofluoric acid (HF) would induce a series of analytical problems, such as special equipment for HF resistant systems, artificial isotope fractionation caused from gaseous silicon loss” (Line 162-176).

3)      Be careful with sentence structure and use of (small) words e.g. line 209 – “silicon is the defining element of silicate reservoirs in Earth” should read “on Earth”. (As an additional aside, I’m also not entirely convinced I understand the point being made in this sentence). E.g. line 274 “with referring to previous study by…” should read “with reference to the previous study by…. E.g. line 359 – “always priority” should read “always the priority” (or probably better written as “preferentially taken up”), and so on.

  According to the suggestive comment, the inappropriate usage of preposition has been corrected (Line 229). The sentence has been revised as “Despite silicon is the defining element of silicate reservoirs on Earth, there is still no clear understanding of how much is hosted in Earth's core, how the silicon enriched continental crust forms from a long history of weathering, erosion and subduction” (Line 229-231).

  The points mentioned by the reviewer have been corrected individually (Line 321, 390-391).

4)      Be careful with labelling and referencing to figures. It’s incredibly difficult to follow in the PDF I have because of the track changes, but I think some of the figures are referenced incorrectly in the text e.g. line 263 refers to figure 4b, but I think this should be figure 3?

  The final clean version will be submitted to the manuscript system, in order to avoid any confusion to the reviewer.

5)      There are a number of spelling mistakes that need fixing e.g. line 404 (polyermized) and line 408 (negatively charged) [sic].

  As the reviewer pointed out, these typos have been corrected. The English grammars and spellings though the whole manuscript has been checked carefully in this revised version.

6)      Be careful with author names e.g. line 406 (should read De La Rocha) and in the caption for figure 8 (should read De La Rocha and Bickle).

  As the reviewer suggested, the author names in the text and figure captions have been double-checked to be correct in the revised version.

7)      Please ‘accept’ all track changes before resubmitting (and be clear as to the links with the supplementary information).

  We are sorry to make reviewers inconvenient to read the manuscript. A clean version without changes tracking will be uploaded to the system. 

Reviewer 2 Report

Overall this seems like a thorough review worthy of publication. It appears that many of the changes I might have suggested have already been made. It does not read as though it was written by a native English speaker, but for the most part it is understandable. A few detailed comments are below.

Line 39 – remove “the”

Line 40 – remove “the”

Figure 1 – Fonts need to be much larger on labels

Figure 2 – Again, much larger fonts on all labels, and the legend. Part A needs to be larger as a whole to be legible- an overall increase in the size of this figure is needed (or separate part A as a separate figure).

Figure 10 – the inset graph is too small to be readable. I’d recommend separating it into a separate figure/subfigure

Author Response

Overall this seems like a thorough review worthy of publication. It appears that many of the changes I might have suggested have already been made. It does not read as though it was written by a native English speaker, but for the most part it is understandable. A few detailed comments are below.

   We are grateful to the reviewer for the suggestive and valuable comments from the three-run of reviewing, which greatly improve the manuscript quality and the substantial discussion and interpretation. In addition, an exhaustive grammatical editing through the whole text has been made by Dr. Simon V. Hohl. All issues related to the grammatical errors raised by the reviewers have been carefully corrected in the revised version.

Line 39 – remove “the”

Line 40 – remove “the”

    The inappropriate usage of article has been corrected through the whole manuscript.

Figure 1 – Fonts need to be much larger on labels

    As the reviewer suggested, the graph has been replotted to present clear illustration.

Figure 2 – Again, much larger fonts on all labels, and the legend. Part A needs to be larger as a whole to be legible- an overall increase in the size of this figure is needed (or separate part A as a separate figure).

   The graph has been modified, in which all labels and the legends have been enlarged and the insert plot (Figure 9b) has been separated to be laid out with Figure 9a vertically.

Figure 10 – the inset graph is too small to be readable. I’d recommend separating it into a separate figure/subfigure

  The insert graph has been enlarged to present better illustration in the revised version.

Round 2

Reviewer 1 Report

This revised manuscript is an improvement, most of my comments have been addressed in a satisfactory way, and I approve of the emphasis on Si complexation and thermodynamics in the review.

My main concern is in section 4.5. I think that the treatment of marine silicon cycling and silicon isotope archives is a little confused. For example, the authors refer to ‘large geological time scales’, and then ‘local scales’ or ‘minor scales’, which seems to me to be mixing up temporal changes (both variability and secular changes) and spatial variability/heterogeneity. This section could treat spatial and temporal ‘scales’ more distinctly to avoid this confusion. Please also edit the subtitle of this section accordingly.

Also, I find section 5.3. is a little out of place and repetitive, and would probably sit better (in a condensed form) in section 4.5.

I also still think that explaining biological uptake as being “well described” by Rayleigh fractionation is an over-simplification (e.g. line 34, section 4.4., line 754), as there are lots of biological systems where this is not the case. Whilst discussed somewhat in section 4.4., this caveat should at least be stated in the abstract and conclusion to avoid misleading the reader.

Furthermore, there are still many structural and grammatical issues that need to be addressed by the editor before I would consider the manuscript ready for publication.

Note that this is not an exhaustive list, and careful editing is still required.

Line 174 – “typical geological process, the silicon isotope geochemistry…” should read “typical geological processes, silicon isotope geochemistry…”.

Line 178 – “in details below” should read “in detail below”.

Line 185 –“Despite silicon is the defining element…” should read “Despite silicon being the defining element…”.

Line 188 – “…the silicon isotope fractionation…” should read “… silicon isotope fractionation…”.

Line 211 – “…, which turns out that…” should read e.g. “…, which revealed that…”.

Line 269 – This sentence is not clear. What is the ‘it’ being referred to? Please rephrase.

Line 306 – ‘preferential enrichments of light isotope…” should read e.g. “preferential enrichments of the light(er) isotope(s)…”.

Line 332 – “Besides, it is consistent that…” should read e.g. “This results is consistent with the observation that…”.

Line 343 – This sentence doesn’t make sense. Please rephrase, and link more coherently with the following sentence.

Line 346 – “occurred” should read “occurs” (there are a few other instances in this paragraph of the use of a past tense, when a present tense would be more appropriate).

Line 353 – “… are that silica substitutes for double-corner…” should read e.g. “results from the substitution of silica for double-corner…”.

Line 372 – “evidences” should read “evidence”.

Line 388 – “prevailing” should read “prevalence”.

Line 397 – This sentence is too long, and should be subdivided.

Line 402 - This sentence doesn’t make sense. Please rephrase.

Line 406 – This sentence is awkwardly phrase. Also, it should be noted that marine biogenic opal can be significantly isotopically lighter than ambient water due to biological fractionation. I think probably the best option is simply to remove “at the end, the precipitated silicon in sea and marine organism is rich in 30Si”.

Line 409 – “fractionations” should read “fractionation”.

Line 410 – “…potential of d30Si proxy for the paleo-reconstruction of…” needs to read either “…potential of the d30Si proxy to be used as an archive of past…” or “…potential use of the d30Si proxy for the paleo-reconstruction…”.

Line 420 – “uptaking” should read “the uptake of”.

Line 425 – “lager” is clearly the wrong word, perhaps “larger”?

Line 425 – “it is as high as 58-76 kg/ha/y”… what does “it” refer to? Please specify.

Line 458 – “organism” should read “organisms”. Please also rephrase the following sentence.

Line 470 – “With comparing the…” should read “By comparing the…”, and “it was proposed that two groups…” should read “it was proposed that the two groups…”.

Line 476 – “recent study” should read “recent studies”. (In fact, I don’t think that this sentence is relevant and could simply be removed).

Line 493 – The sentence starting “At the other hand…” is rather confused, and needs rephrasing. I’m not really sure what is being communicated here.

Line 501 – remove “Besides” from the beginning of this sentence.

Line 502 – “concentration the” should read “concentration of the”.

Line 505 – “bio-siliceous” should read e.g. “bio-siliceous materials”.

Line 507 – What is “It” at the beginning of this sentence?

Line 510 – remove “nutrition” from this sentence.

Line 513 – “diatom and sponge of d30Si as a…” should read “diatom and sponge d30Si as a…”. Note also that sponge d30Si is not technically a proxy for DSi utilisation, but more related to the ambient DSi concentration in bottom waters.

Line 544 – “comparison on” should read “comparison of”.

Line 548 – “possible hypotheses to the apparent contradiction…” should read “possible hypotheses to explain the apparent contradiction…”.

Line 584 – What is meant by reported “individually”? Do you simply mean that different studies are consistent? Please clarify.

Line 598 - This sentence is too long, and should be subdivided.

Line 611 – “explored” should read “found”.

Line 657 – I think that “implication” is the wrong word here, perhaps consider rephrasing to e.g. “Silicon isotopic measurements used in combination with other geochemical indices… can be used to better constrain the genesis of ore deposits.”.

Line 665 onwards. This section needs restructuring to clearly distinguish the two hypotheses for ore formation using correct punctuation.

Line 668 – “behaves” should read e.g. “is characterized by” or “exhibits” etc.

Line 671 – “radiolarian” should read “radiolaria”.

Line 695 – What is “It” at the beginning of this sentence? Please rephrase.

Line 703 – “with the d30Si characters” should read “using the d30Si composition” or “using the d30Si characteristics”.

Line 709 – “In hydrosphere” should read “In the hydrosphere”, and “evaluating” should read either “in evaluating” or “evaluate”.

Line 711 – “Combination” should read “In combination”.

Line 712 – “plant” should read “plants”.

Line 713 - “pioneer” should read “pioneering”

Line 718 – “the mixing process” should read “mixing processes”.

Line 725 - “silicon isotope in the ocean is not homogeneous” should read e.g. “silicon isotopic compositions of marine waters are not homogeneous”.

Line 731 – ‘hydrosphere” should read “the hydrosphere”.

Line 751-752 - “chemical weathering process” should read “chemical weathering processes”.

Figures:

Whilst figure 9 has been improved, I still don’t think it’s as clear as it could be. Consider using hollow and solid symbols (rather than smaller half-symbols) to distinguish DSi and d30Si?

Author Response

Reviewer’s original comments are in italic fonts and black

Authors’ reply and revisions are in regular font and blue/red

Reviewer 1#:

This revised manuscript is an improvement, most of my comments have been addressed in a satisfactory way, and I approve of the emphasis on Si complexation and thermodynamics in the review.

My main concern is in section 4.5. I think that the treatment of marine silicon cycling and silicon isotope archives is a little confused. For example, the authors refer to ‘large geological time scales’, and then ‘local scales’ or ‘minor scales’, which seems to me to be mixing up temporal changes (both variability and secular changes) and spatial variability/heterogeneity. This section could treat spatial and temporal ‘scales’ more distinctly to avoid this confusion. Please also edit the subtitle of this section accordingly.

Also, I find section 5.3. is a little out of place and repetitive, and would probably sit better (in a condensed form) in section 4.5.

Response: According to the suggestive comment, the section 5.3 was combined into the section 4.5, in which relevant repetitive parts have been deleted (Line 485-555). The second-level and the third-level of subtitles were revised as below: 4.5. Silicon cycling in hydrosphere; 4.5.1. Global spatial and temporal variability/heterogeneity in oceanic Si cycle and δ30Si; 4.5.2. δ30Si variation in terrestrial hydrosphere.

I also still think that explaining biological uptake as being “well described” by Rayleigh fractionation is an over-simplification (e.g. line 34, section 4.4., line 754), as there are lots of biological systems where this is not the case. Whilst discussed somewhat in section 4.4., this caveat should at least be stated in the abstract and conclusion to avoid misleading the reader.

Response: As the reviewer suggested, the relevant interpretations on the biological fractionations of silicon isotopes have been modified in the individual sections:

(1)  Section 4.4.: “The role of complexation in biological fractionations of silicon isotopes is very poorly understood, largely because the complicated biochemical pathways, likely involving several enzymatic processes and active transport proteins, are not well constrained in any siliceous organism” (Line 451-454); “Both De La Roach et al. and our previous work presented comprehensive studies on silicon isotope dynamic fractionation in solution and organism [16, 83], where the process of silicon isotope dynamic fractionation is consistent with the Rayleigh fractionation model as same as that in silica precipitation [16, 108]” (Line 459-462); “In addition to the Rayleigh effect in the dynamic fractionation of silicon isotope during the biological uptake into various plants, other controlling factors should be considered to explain obvious differences in both δ30Si values and fractions of dissolved H4SiO4 from roots to stem and leaves [22, 23] ” (Line 469-472).

(2)  Abstract Section: “The role of complexation in biological fractionations of silicon isotopes is more complicated, likely involving several enzymatic processes and active transport proteins” (Line 34-36).

(3)  Conclusion Section: “(iv) In the biological uptake processes, biological fractionations of silicon isotopes are very poorly understood, largely because the complicated biochemical pathways, likely involving several enzymatic processes and active transport proteins” (Line 752-754).

Furthermore, there are still many structural and grammatical issues that need to be addressed by the editor before I would consider the manuscript ready for publication.

Note that this is not an exhaustive list, and careful editing is still required.

(1)Line 174 – “typical geological process, the silicon isotope geochemistry…” should read “typical geological processes, silicon isotope geochemistry…”.

Response: The sentence has been revised as “With further understanding on the driving mechanisms for silicon isotope fractionations behind typical geological processes” (Line 174-175).

(2)Line 178 – “in details below” should read “in detail below”.

 Response: The sentence has been revised as “which will be discussed in detail below.” (Line 178-179).

(4)  Line 185 –“Despite silicon is the defining element…” should read “Despite silicon being the defining element…”.

Response: This sentence has been revised as “Despite silicon is being the defining element of silicate reservoirs on Earth, there is still no clear understanding of how much is hosted in Earth's core, how the silicon enriched continental crust forms from a long history of weathering, erosion and subduction [59].” (Line 186-188).

(4)Line 188 – “…the silicon isotope fractionation…” should read “… silicon isotope fractionation…”.

Response: The sentence has been revised as “Like that of any other elements, silicon isotope fractionation is relatively limited at high temperatures [84].” (Line 188-189).

(5)Line 211 – “…, which turns out that…” should read e.g. “…, which revealed that…”.

Response: This sentence has been revised as “which revealed out that the equilibrium silicon isotope fractionation trend is not directly connected to the polymerization degree.” (Line 212-213).

(5)  Line 269 – This sentence is not clear. What is the ‘it’ being referred to? Please rephrase.

 Response: According to the suggestive comment, it means the equilibrium solubility. This sentence has been revised as “Generally, the equilibrium solubility of Si increases with increasing of environmental temperatures, and is highly variable with variations in temperature, pressure, and differences in specific surface area and Al content of bio-siliceous fragments [96].” (Line 270-272).

(7)Line 306 – ‘preferential enrichments of light isotope…” should read e.g. “preferential enrichments of the light(er) isotope(s)…”.

Response: This sentence has been revised as “Similar preferential enrichments of lighter isotopes in the solid phase also have been observed, such as for Fe-isotopes during rapid precipitation of hematite [110].” (Line 306-308).

(8)Line 332 – “Besides, it is consistent that…” should read e.g. “This results is consistent with the observation that…”.

Response: This sentence has been revised as “Besides, the results are consistent the observation that silicon isotope fractionation during silica precipitation is temperature dependent and more significant at low temperature [38, 108-109].” (Line 333-335).

(9)Line 343 – This sentence doesn’t make sense. Please rephrase, and link more coherently with the following sentence.

 Response: This sentence has been revised as “During chemical weathering processes (Reaction I), silicon either is being released into continental surface and ground waters, or transported into soils by uptake of plants, and formation of secondary precipitates (e.g. metastable silica-containing solids), as well as adsorption onto secondary oxides. The associated dynamic fractionation occurs in precipitation during the weathering of silicate minerals” (Line 344-348).

(10)Line 346 – “occurred” should read “occurs” (there are a few other instances in this paragraph of the use of a past tense, when a present tense would be more appropriate).

 Response: This sentence has been revised as “The associated dynamic fractionation occurs in precipitation” (Line 347).

(11)Line 353 – “… are that silica substitutes for double-corner…” should read e.g. “results from the substitution of silica for double-corner…”.

 Response: The sentence has been revised as “Results from the substitution of silica for double-corner FeO6-octahedra in iron oxy-hydroxide polymeric complexes existing at the early stages of Fe(III) hydrolysis, silica substitutes for double-corner likely by forming 2C-type (double corner) complexes with small Fe oxy-hydroxide polymers whose structure consists of FeO6-octahedra linked together by common edges [121]..” (Line 352-356).

(12)Line 372 – “evidences” should read “evidence”.

 Response: This word has been corrected as “evidence (Line 371).

(13)Line 388 – “prevailing” should read “prevalence”.

 Response: The sentence has been revised as “With the prevalence of silicon isotope as a useful weathering proxy, it raised the question of the role of mineralogy on silicon isotope fractionation by silicon incorporation into secondary phases.” (Line 388-389).

(14)Line 397 – This sentence is too long, and should be subdivided.

Response: According to the suggestive comment, this sentence has been divided into two short ones “This study clearly indicated that the silicon isotope composition in secondary clay minerals was controlled by the degree of soil desilication, which was related to the rainfall pattern governing the formation of second minerals. Silicon isotope fractionation between the parent silicate material and the secondary clay minerals increases following the order of smectite, allophane, halloysite and kaolinite, which corresponds to a smaller isotope fractionation for Si-rich clay minerals (e.g. smectite) and a larger isotope fractionation for Si-poor clay minerals such as kaolinite [13].” (Line 396-402).

(15)Line 402 - This sentence doesn’t make sense. Please rephrase.

 Response: According to the suggestive comment, this sentence “It well explained the variation trend of δ30Si in secondary clay minerals, such as slightly lighter δ30Si in smectite (-0.16 to -0.52 ‰) [52] and very much lighter in kaolinite (-2.2‰) [26, 126], compared to the limited spread of δ30Si (i.e. -0.29 ± 0.08 ‰) in silicate rocks (BSE) [3, 58, 129]” (Line 402-405).

(16)Line 406 – This sentence is awkwardly phrase. Also, it should be noted that marine biogenic opal can be significantly isotopically lighter than ambient water due to biological fractionation. I think probably the best option is simply to remove “at the end, the precipitated silicon in sea and marine organism is rich in 30Si”.

 Response: According to the suggestive comment, this sentence has been revised as “In summary, secondary minerals formed with high Al/Si ratios, are generally depleted in 30Si (see [129-130]), and those in river water are enriched in 30Si over the host rock (e.g. [50, 125-128]). In addition, the δ30Si values show an upward trend from upstream to downstream [4]. The associated silicon isotope fractionations during the crustal surficial weathering are schematically shown in Figure 6” (Line 406-410).

(17)Line 409 – “fractionations” should read “fractionation”.

Response: It has been corrected in the revised text (Line 409).

(18)Line 410 – “…potential of d30Si proxy for the paleo-reconstruction of…” needs to read either “…potential of the d30Si proxy to be used as an archive of past…” or “…potential use of the d30Si proxy for the paleo-reconstruction…”.

 Response: The sentence has been revised as “The integrated understanding greatly strengthens the potential use of δ30Si proxy for the paleo-reconstruction of climatic conditions in soil weathering environments” (Line 410-411).

(19)Line 420 – “uptaking” should read “the uptake of”.

 Response: The part has been corrected as “the uptake of” (Line 420).

 (20)Line 425 – “lager” is clearly the wrong word, perhaps “larger”?

 Response: The typo has been corrected (Line 425).

(21)Line 425 – “it is as high as 58-76 kg/ha/y”… what does “it” refer to? Please specify.

 Response: It means the contribution of plants to biogeochemical Si. This sentence has been revised as “For instance, the contribution of plants to biogeochemical Si is as high as 58-76 kg/ha/y in the equatorial forest ecosystems [133].” (Line 425-426).

(22)Line 458 – “organism” should read “organisms”. Please also rephrase the following sentence.

Response: The word appeared in the whole text has been corrected as “organisms”.

 (23)Line 470 – “With comparing the…” should read “By comparing the…”, and “it was proposed that two groups…” should read “it was proposed that the two groups…”.

Response: This sentence has been revised as “By comparing the Si isotope fractionation observed during the formation of sponge spicules and in silicification of diatom opal [19-20], it was proposed that the two groups evolved Si acquisition mechanisms with different impacts on silicon isotopes [25].” (Line 470-473).

(24)Line 476 – “recent study” should read “recent studies”. (In fact, I don’t think that this sentence is relevant and could simply be removed).

Response: This sentence has been revised as “The most recent study proposed that sponges have initiated seawater oxygenation by redistributing organic carbon oxidation through filtering suspended organic matter from seawater, and the resulting increase in dissolved oxygen levels potentially triggered the diversification of eumetazoans [147]” (Line 476-478). It is the first work to link the silicon isotopes to the ocean oxygenation, , and has been kept in the revised version.

(25)Line 493 – The sentence starting “At the other hand…” is rather confused, and needs rephrasing. I’m not really sure what is being communicated here.

Response: This sentence has been revised as “The evolution of seawater δ30Si over geological time was derived from the recorded δ30Si in cherts, radiolarian and sponge spicules, diatoms (Figure 8c and 8d), which shows a rise trend compared to the Archean ocean generally.” (Line 493-495).

(26)Line 501 – remove “Besides” from the beginning of this sentence.

Response: It has been deleted (Line 501).

(27)Line 502 – “concentration the” should read “concentration of the”.

Response: This sentence has been revised as “In general, H4SiO40 concentration of the surface is the lowest and increases from the surface to 1000 m (Figure 9a) [24-25; 149-150].” (Line 501-502).

(28)Line 505 – “bio-siliceous” should read e.g. “bio-siliceous materials”.

Response: This sentence has been revised as “the silica solubility of surficial bio-siliceous materials (e.g. Southern Ocean sediments, plankton, culture diatoms, and amorphous silica gel) as a function of temperature was plotted in Figure 9b.” (Line 504-506).

(29)Line 507 – What is “It” at the beginning of this sentence?

Response: This sentence has been revised as “The equilibrium silica solubility increases with increasing of environmental temperature.” (Line 506-507).

(30)Line 510 – remove “nutrition” from this sentence.

Response: According to the suggestive comment, the “nutrition” has been removed. (Line 510).

(31)Line 513 – “diatom and sponge of d30Si as a…” should read “diatom and sponge d30Si as a…”. Note also that sponge d30Si is not technically a proxy for DSi utilisation, but more related to the ambient DSi concentration in bottom waters.

 Response: This sentence has been revised as “Therefore, the relationships between H4SiO40 and δ30Si in sponge spicules and diatoms are highly supportive of the qualitative use of diatom and sponge δ30Si as a proxy for the ambient DSi concentration in bottom water” (Line 511-513).

(32)Line 544 – “comparison on” should read “comparison of”.

 Response: This sentence has been revised as “The comparison on of published δ30Si data relative to NBS-28 for samples from planetary bodies in the inner solar system concluded that the lunar mantle and BSE have similar δ30Si.” (Line 566-567).

(33)Line 548 – “possible hypotheses to the apparent contradiction…” should read “possible hypotheses to explain the apparent contradiction…”.

Response: This sentence has been revised as “The study proposed two possible hypotheses to explain the apparent contradiction as:” (Line 569-570).

(34)Line 584 – What is meant by reported “individually”? Do you simply mean that different studies are consistent? Please clarify.

Response: This sentence has been revised as “Relatively consistent Si-core concentrations of 13 wt. % [51], 6 wt. % [6], 1 wt. % [54-55], and 12 wt. % [8] are reported.” (Line 604-606).

(35)Line 598 - This sentence is too long, and should be subdivided.

Response: According to the suggestive comment, this sentence has been revised as “Recently, the equilibrium isotope fractionation during the high-temperature partitioning of stable isotopes of rock-forming elements (e.g. Mg, Si, Fe, H, O) has been reviewed by Young et al [169]. The limiting parameters of using silicon isotopes to constrain the conditions for core formation are the uncertainty in D30SiBSE-BE because disagreement about the typical 30Si/28Si of chondrites and the inevitable uncertainty as to whether bulk Earth is truly chondritic in silicon isotope rather than the fractionation factor” (Line 620-625).

(36)Line 611 – “explored” should read “found”.

Response: The sentence has been revised as “It has been explored found that chert within BIFs exhibits a largely negative silicon isotope signature ranging from about -2.5‰ to -0.5‰ in δ30Si which has been interpreted as a hydrothermal signal [10, 33, 171-172].” (Line 632-634).

(37)Line 657 – I think that “implication” is the wrong word here, perhaps consider rephrasing to e.g. “Silicon isotopic measurements used in combination with other geochemical indices… can be used to better constrain the genesis of ore deposits.”.

Response: The sentence has been revised as “Silicon isotopic measurements used in combination with other geochemical indices (e.g. δ18O and δ11B) can be used to better constrain the genesis of ore deposits.” (Line 677-678).

(38)Line 665 onwards. This section needs restructuring to clearly distinguish the two hypotheses for ore formation using correct punctuation.

Response: The correct punctuation has been used. (Line 687-699).

(39)Line 668 – “behaves” should read e.g. “is characterized by” or “exhibits” etc.

Response: This sentence has been revised as “It is widely distributed with larger thickness, and behaves exhibits higher δ30Si (+0.2 ~ +0.8‰) and wide variation of δ18O values (+12.0 ~ +21.4‰).” (Line 687-689).

(40)Line 671 – “radiolarian” should read “radiolaria”.

Response: The sentence has been revised as “The existence of large amount of radiolaria in these rocks further approved the biochemical deposition genesis.” (Line 691-692).

(41)Line 695 – What is “It” at the beginning of this sentence? Please rephrase.

Response: This sentence has been revised as “Clay minerals from different sources can be distinguished by the δ30Si characters (Figure 11).” (Line 716-717).

(42)Line 703 – “with the d30Si characters” should read “using the d30Si composition” or “using the d30Si characteristics”.

Response: This sentence has been revised as “Therefore, it is possible to constrain the origins of clay minerals in accordance using the δ30Si characters [10].” (Line 724-725).

(43)Line 709 – “In hydrosphere” should read “In the hydrosphere”, and “evaluating” should read either “in evaluating” or “evaluate”.

Response: This sentence has been revised as “the variation of δ30Si could help us evaluate the silicon isotopic budgets in terrestrial aqueous systems.” (Line 541-542).

(44)Line 711 – “Combination” should read “In combination”.

Response: According to the suggestive comment, this sentence has been revised as “In combination with the silicon isotopes in rock, fluids (e.g. lake, groundwater, river, and ocean).” (Line 722-723).

(45)Line 712 – “plant” should read “plants”.

Response: This sentence has been deleted.

(46)Line 713 - “pioneer” should read “pioneering”

Response: This sentence has been revised as “After the pioneering works on the δ30Si of dissolved silicic acid in oceans.” (Line 542-543).

(47)Line 718 – “the mixing process” should read “mixing processes”.

Response: This sentence has been revised as “Besides, the seasonal variation, the weathering degree and mixing process also affect the silicon isotope compositions in surface water.” (Line 547-548).

(48)Line 725 - “silicon isotope in the ocean is not homogeneous” should read e.g. “silicon isotopic compositions of marine waters are not homogeneous”.

Response: This sentence has been revised as “The silicon isotope compositions of marine waters are not homogeneous.” (Line 513-514).

(49)Line 731 – ‘hydrosphere” should read “the hydrosphere”.

Response: This part has been combined into Section 4.5., and the repetitive statement has been deleted in the revised version.

(50)Line 751-752 - “chemical weathering process” should read “chemical weathering processes”.

Response: This sentence has been revised as “chemical weathering processes, the isotope fractionation is controlled by the associated dynamic fractionation in precipitation during the weathering of silicate minerals, the selective adsorption of H4SiO40 on soil Fe-oxides/Al-oxides, the degree of soil desilication.” (Line 747-750).

Figures:

Whilst figure 9 has been improved, I still don’t think it’s as clear as it could be. Consider using hollow and solid symbols (rather than smaller half-symbols) to distinguish DSi and d30Si?

Response:  As the reviewer suggested, the Figure 9 has been modified. The hollow and solid symbols were used to represent DSi and δ30Si, respectively, which gives better illustration of this graph.

Additional Changes:

1.     In Figure 1, a typo of Tourmalites” has been corrected as “Tourmalinites” and the graph has been replotted.

2.     The spelling and grammatical issues through the whole text have been carefully checked in the revised version.